# 🏴 Quilt-1M: One Million Image-Text Pairs for Histopathology

**Wisdom O. Ikezogwo**[*]🏴   **Mehmet S. Seyfioglu** 🏴   **Fatemeh Ghezloo** 🏴
**Dylan Geva**   **Fatwir S. Mohammed**   **Pavan K. Anand**
**Ranjay Krishna**   **Linda G. Shapiro**
University of Washington
{wisdomik,msaygin,fghezloo,dgeva,pka4,ranjay,shapiro}@cs.washington.edu
fatwir@uw.edu

## Abstract

Recent accelerations in multi-modal applications have been made possible with the plethora of image and text data available online. However, the scarcity of analogous data in the medical field, specifically in histopathology, has slowed comparable progress. To enable similar representation learning for histopathology, we turn to YouTube, an untapped resource of videos, offering $1,087$ hours of valuable educational histopathology videos from expert clinicians. From YouTube, we curate QUILT: a large-scale vision-language dataset consisting of $802,144$ image and text pairs. QUILT was automatically curated using a mixture of models, including large language models, handcrafted algorithms, human knowledge databases, and automatic speech recognition. In comparison, the most comprehensive datasets curated for histopathology amass only around 200K samples. We combine QUILT with datasets from other sources, including Twitter, research papers, and the internet in general, to create an even larger dataset: QUILT-1M, with 1M paired image-text samples, marking it as the largest vision-language histopathology dataset to date. We demonstrate the value of QUILT-1M by fine-tuning a pre-trained CLIP model. Our model outperforms state-of-the-art models on both zero-shot and linear probing tasks for classifying new histopathology images across 13 diverse patch-level datasets of 8 different sub-pathologies and cross-modal retrieval tasks[2].

## 1   Introduction

Whole-slide histopathology images are dense in information, and even individual image patches can hold unique, complex patterns critical for tissue characterization. Summarizing this information into a single label is an oversimplification that fails to capture the complexity of the field, which covers thousands of evolving disease sub-types [55]. This highlights the need for more expressive, dense, interconnected representations beyond the reach of a singular categorical label. As such, natural language descriptions can provide this comprehensive signal, linking diverse features of histopathology sub-patch structures [20, 24].

If there were a large-scale vision-language dataset for histopathology, researchers would be able to leverage the significant advancements in self-supervised vision and language pre-training to develop effective histopathology models [46]. Unfortunately, there is a significant scarcity of comprehensive datasets for histopathology. Notable open-source contributions have been made with datasets like ARCH [20] and OpenPath [24]. Yet, these sources are still somewhat limited due to their size, as the

---

[*]Reach corresponding author at wisdomik@cs.washington.edu; 🏴: Equal contribution.
[2]The data and code will be available at Quilt-1M

37th Conference on Neural Information Processing Systems (NeurIPS 2023) Track on Datasets and Benchmarks.

former has only $\approx 8$K samples and the latter (the largest histopathology vision-language dataset to date) has about 200K samples. Although recent efforts (e.g. PMC-15M [67]) curated 15M image-text pairs across a variety of different biomedical domains from Pubmed [48], whether their samples are specific to histopathology remains ambiguous; worse, their dataset is not openly available.

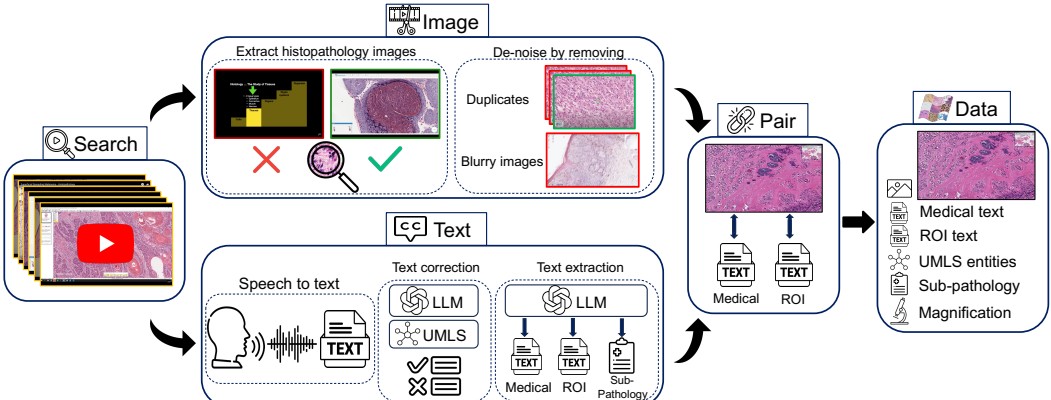

Figure 1: **Overview of QUILT curation pipeline.** We identify relevant histopathology YouTube videos in **Search**. For **Image** extraction, we find and de-noise histopathology frames using trained models. In **Text** section, we rely on a conventional Automatic Speech Recognition (ASR) model and leverage Unified Medical Language System (UMLS) and large language models (LLMs) for post-processing and ASR error correction. Relevant sub-pathology, medical and region-of-interest (ROI) text are extracted using an LLM. Finally, domain-specific algorithms are used to **Pair** images and text, eliminating duplicates to yield QUILT, a richly annotated image-text dataset for histopathology.

To address the need for a large-scale vision-language dataset in histopathology, we introduce QUILT: containing $437,878$ images aligned with $802,144$ text pairs across multiple microscopic magnification scales covering from 10x to 40x. We draw on the insight that publicly available educational YouTube histopathology content represents an untapped potential. We curate QUILT using $1,087$ hours of valuable educational histopathology videos from expert pathologists on YouTube. To extract aligned image and text pairs from the videos, we utilize a mixture of models: large language models (GPT-3.5), handcrafted algorithms, human knowledge databases, and automatic speech recognition. QUILT does not overlap with any current open-access histopathology data sources. This allows us to merge our dataset with other open-source datasets available. Therefore, to create an even larger and more diverse dataset, we combine QUILT with data from other sources, such as Twitter, research papers, and the Internet, resulting in QUILT-1M. The larger QUILT-1M contains one million image-text pairs, making it the largest public vision-language histopathology dataset to date.

Using QUILT and QUILT-1M, we finetune vision-language models using a contrastive objective between the two modalities. We extensively evaluate it on 13 external histopathology datasets taken across different sub-pathologies. We report zero-shot classification, linear probe, and image-to-text and text-to-image retrieval tasks. Against multiple recently proposed baselines (CLIP [46], PLIP [24], and BiomedCLIP [67]), models trained with QUILT-1M outperform all others. Our ablations identify the importance of QUILT.

QUILT offers three significant advantages: First, QUILT does not overlap with existing data sources; it ensures a unique contribution to the pool of available histopathology knowledge. Second, its rich textual descriptions extracted from experts narrating within educational videos provide more expressive, dense interconnected information. Last, the presence of multiple sentences per image fosters diverse perspectives and a comprehensive understanding of each histopathological image. We hope that both computer scientists and histopathologists will benefit from QUILT's potential.

## 2   Related work

We built upon a growing literature applying self-supervised learning and other machine learning methods to medical image understanding.

**Machine learning for histopathology.** Early representation learning work in computational pathology primarily relied on weakly-supervised learning, with each whole-slide image (WSI) receiving a single label. The limited nature (single label to many patches) has produced sub-optimal models [12, 26] at the patch level. Lately, a self-supervised learning approach, which learns useful representations from unlabeled data, has shown some success [26, 13, 12]. Most of this work has been unimodal. They use image augmentations similar to those used for natural images [14], mostly differing by way of consciously injecting domain knowledge. For example, they leverage the compositional nature of H&E stain information of whole-slice images [26], or inject hierarchical morphological information at different magnifications [13], or combine with other modalities like genomic features [12] or with descriptive text [20]. When text data is used, the objectives similarly use augmentations seen in natural language [50]. By contrast, we explore self-supervised mechanisms that learn better histopathology information representations that go beyond a single label, aided by language descriptions.

**Medical vision-language datasets.** Learning vision-language representations demands a large dataset of images aligned with descriptive text, a resource that is notably lacking in histopathology. The MIMIC-CXR-JPG v2.0.0 dataset [28], for example, consists of de-identified hospital-sourced chest radiographs and reports. For histopathology, The Cancer Genome Atlas[3] provides de-identified PDF-reports for a limited number of WSIs. Despite this resource, the enormous size of this data (reaching up to $120,000^2$ pixels) makes processing challenging, limiting its use to a small number of focused studies [39]. A majority of medical vision-language datasets are concentrated in the radiology sub-domain, due to the relatively straightforward process of collecting validated multimodal data [28]. Many models are trained on a subset of PubMed [48] or comparable radiology datasets [68, 23, 18, 43]. PMC-15M [67], a recent subset of PubMed not specific to histopathology, was used to train multiple models. While the models themselves are public, PMC-15M is not, making it hard to determine what portion of it is histopathology-relevant.

**Vision-language pairs on histopathology.** One of the first histopathology vision-language datasets, ARCH, contains only $7,614$ accessible image-text pairs [20, 22]. Later on, [24] released OpenPath, a dataset of 200K image-text pairs extracted from Twitter. This was the largest histopathology dataset until QUILT-1M.

**Video data for self-supervision.** Numerous recent studies have started to tap into video data. For instance, millions of publicly accessible YouTube videos were used to train a vision-language model [65, 66]. Similarly, a causal video model was trained by using sequential gaming videos [6]. Localized narratives [58, 44] provide another example of dense, interconnected supervision for a single image. Despite the potential of video content, video often yields noisier datasets compared to static sources. Recently, the enhanced capabilities of automatic speech recognition models streamlined the curation of large-scale cleaner datasets from videos [65, 6, 67]. Furthermore, the growing versatility of large language models has shown promise as data annotators, information extractors [33, 59, 15, 21], text correctors [63], and as tools for medical information extraction and reasoning [1, 56].

## 3    Curating QUILT: Overview

Creating a vision-language dataset from videos is a significant undertaking, as not all videos are suitable for our pipeline. Many either lack voiced audio, are not in English, fail to contain medically relevant content, or have insufficient medical relevance—for example, videos that present static images of histopathology content on a slide deck, or those that briefly cover histopathology images in pursuit of a different objective. Conventional automatic speech recognition (ASR) systems also struggle with the specialized requirements of histopathology transcription, necessitating a non-trivial solution. The de-noising of text and image modalities adds further complexity as the videos are typically conversational and, therefore, inherently noisy. Instructors pan and zoom at varying speeds, recording a mix of relevant and irrelevant histopathological visual content in their videos. As such, trivially extracting frames at static intervals fails to capture the data appropriately. To collect QUILT we trained models and handcrafted algorithms that leverage the nuances in the instructors' textual and visual behavior, ensuring accurate collection and alignment of both modalities.

---

[3]https://www.cancer.gov/tcga

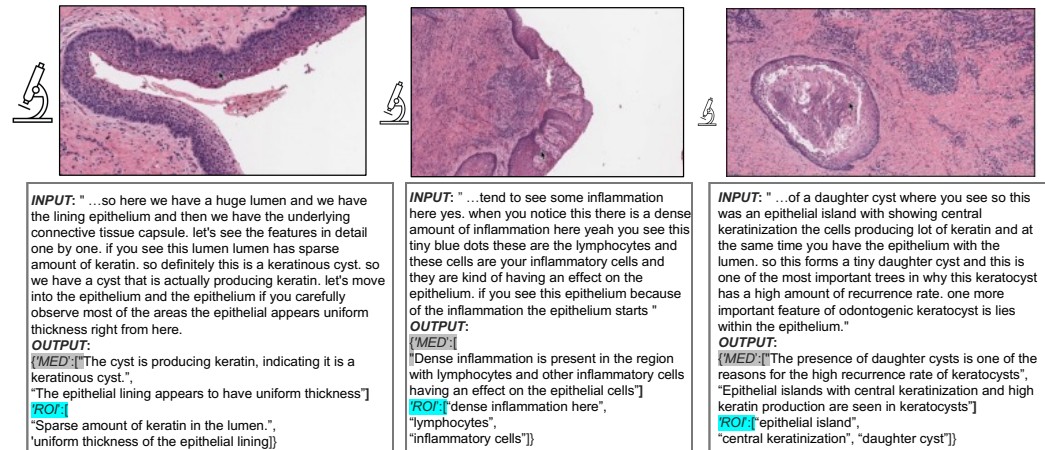

Figure 2: **QUILT examples**. **Input** is the corrected ASR caption for the representative image. **Output** are the medical and ROI extracted text(s) paired with the image (see Section 3.1). In histopathology, understanding tissue characteristics often involves views from varying magnification levels. Thus, in QUILT we estimate an image's magnification (indicated by the relative size of the microscope icon).

## 3.1 QUILT: Collecting medical image and text pairs from YouTube

Our proposed dataset curation pipeline involves (1) gathering channel and video data covering the histopathology domain, (2) filtering videos based on a certain "narrative style", (3) extracting and denoising image and text modalities from videos using various models, tools, and algorithms, (4) postprocessing denoised text by LLMs to extract medical text and finally, (5) splitting and aligning all modalities for curating the final vision-language pre-training (VLP) data. See Figure 1 (and supplemental material) for a detailed overview of the pipeline.

**Collecting representative channels and videos.** Our pipeline begins by searching for relevant channels and video ids on YouTube, focusing on the domain of histopathology. Using keywords spanning 18 sub-pathology fields (see supplement for more details), we search among channels before searching for videos to expedite discovery, considering that video searches are time-consuming and the APIs pose limitations on numerous requests [65]. Channels with subscriber count $\geq 300K$ are excluded to avoid large general science channels, as educational histopathology channels often have fewer subscribers. We then download low-resolution versions of all identified videos, with the lowest resolution at 320p.

**Filtering for narrative-style medical videos.** For each video within each channel, we exclude videos that are shorter than 1 minute, non-voiced, or have non-English audio. For videos meeting these heuristics, two decisions are made:

(A) Do they have the required medical content, i.e., histopathology image-text pairs?

(B) If so, are they in narrative style – videos wherein the presenter(s) spend a significant time panning and zooming on the WSI, while providing vocal descriptions of image content?

For **(A)** we automatically identify the relevant videos by extracting keyframes from a video. These keyframes are automatically extracted using FFmpeg [4], marking the beginning or end of a scene (frames containing significant visual changes). The software requires a threshold that determines the minimum amount of visual change required to trigger a keyframe. Through experimentation, we set different thresholds for various video durations, with smaller thresholds for longer videos. Next, we train and use an ensemble of three histopathology image classifiers to identify videos with histopathology images (See supplement for more details).

For **(B)**, in which we identify narrative-style videos, we randomly select keyframes predicted to be histopathology. For each such selected frame, we extract the next three histopathology key-frames and compute the cosine similarity between the selected frame and each of the subsequent three

---

[4]https://ffmpeg.org/

frames. If all three have similarity scores $\geq$ a preset threshold of $0.9$, we count it as a narrative streak. A video is identified as narrative style if at least 10% of the selected frames exhibit a narrative streak. Consequently, we download all narrative-style videos at high-resolution. Narrative-style videos typically cover WSIs at various magnifications, hence, we train a tissue-image-magnification classifier to predict the following three scales: $\{(1 - 10)\text{x}, (> 10 - 20)\text{x}, (> 20)\text{x}\}$. This provides relevant metadata for downstream objectives.

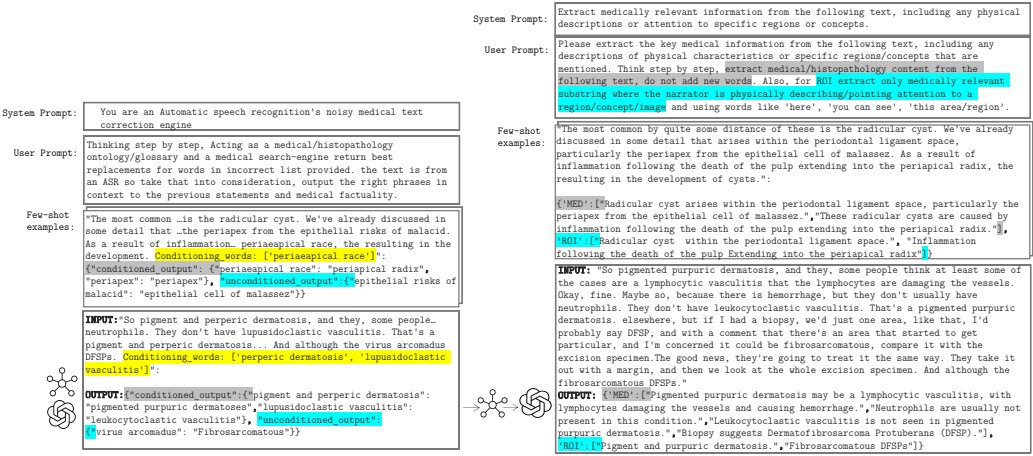

Figure 3: Prompting the LLM by providing few-shot examples to perform the following tasks: (Left) correcting noisy ASR text within its context. We highlight the probable list of misspelled keywords in yellow and their corrections by the LLM in gray. Additional missed errors/misspelled entries identified by the LLM are highlighted in blue. (Right) extracting medical (MED) and ROI text from a given text. We highlight the definition of medical and ROI text in blue and gray respectively.

**Text Extraction using ASR and text denoising.** The high costs associated with private medical ASR APIs [5] necessitated the use of a more conventional ASR model: Whisper [47]. As anticipated, this model often misinterprets medical terms, thus requiring the use of post-processing algorithms to minimize its error rates.

We propose a four-step text de-noising and quality control pipeline: **i)** We utilize the Rake keyword extraction algorithm to extract keywords or key-phrases up to four words and refine them by eliminating stopwords [49]. **ii)** We then cross-check each refined entry against UMLS [7] using the SciSpacy entity linking package [41]. If an entry is not found within UMLS, we check for misspelled words within the entry using a spell-checking algorithm[6], instantiated with a specialized list of histopathology terms curated from various histopathology ontology labels and definitions. **iii)** With this probable list of misspelled keywords, we *condition* and prompt the LLM with examples to correct the misspelled entry within its context (sentence), and secondly, we task the LLM with identifying additional *unconditioned* errors/misspelled entries. For both, we leverage a set of manually curated examples to prompt the LLM in-context Figure 3. For more examples and failure cases, see supplement for more details. **iv)** Finally, to de-noise the text, we resolve the output mapping of incorrect $\mapsto$ correct entries by verifying the corrected words against UMLS and our curated list of histopathology words/phrases. Entries that pass this double-validation process are used to replace the initial noisy transcription. Leveraging domain-specific databases to extract the text and filter out noise allows us to bypass the correction of repetition errors and filler words, such as *'ah', 'uhm', 'the', etc.* in tandem, using LLMs allows us to concentrate on correcting medically-relevant misspelled words, rather than correcting non-medically-relevant terms.

From the ASR-corrected text, we extract *medical text* which describes the image(s) as a whole. Also, when the speaker describes/gestures at visual regions-of-interest through statements like *"look here ..."*, we extract the text entity being described as *ROI text*. To filter relevant medical text and ROI text from the ASR-corrected text, we utilize LLMs (See supplement for more details), a decision rooted in

---

[5]nuance.com/en-au/healthcare/provider-solutions/speech-recognition/dragon-medical-one.html
[6]https://github.com/barrust/pyspellchecker

a few compelling reasons: 1) Curating pre-training datasets at a scale that can tolerate higher levels of noise, LLMs are more cost-effective than expert-human (medical) labor. 2) The task does not require LLMs to generate new information but instead they discriminate useful versus irrelevant signals, serving to improve the signal-to-noise ratio of the data. To extract relevant text, we prompt LLMs to filter out all non-medically relevant text, providing context as necessary. See Figure 2 for some example image-text pairs. Lastly, we instruct the LLMs to refrain from introducing any new words beyond the corrected noisy text and set the model's temperature to zero. Finally, we use LLMs to categorize our videos into one of the 18 identified sub-pathology classes. Similar to the previous tasks, this categorization is done by conditioning with a few examples and prompting the LLM to predict the top three possible classes given the text. More details, prompts, and additional examples are presented in the supplemental material.

**Image frame extraction and denoising.** For each video, we employ a similar method to that described in **Filtering for narrative-style medical videos** subsection to extract histopathology key-frames; our method leverages these frames' times $t$ as beacons to break the entire video into time-intervals called *chunks* from which to extract representative image(s). Next, we extract the median image (pixel-space) of stable (static) frames in each chunk if they exists, else we de-duplicate the histopathology keyframes (beacons of the chunk). In essence, we use the extracted histopathology scene frames as guides for data collection, exploiting the human tendency in educational videos to pause narration during explanation, and we extract the relevant frame(s).

**Aligning both modalities.** For each narrative-style video, we perform the following steps to align image and text modalities: First, we compute histopathology time chunks denoted as $[(t_1, t_2), (t_3, t_4), \cdots (t_{n-1}, t_n)]$ from keyframes after discriminating histopathology frames using the histopathology ensemble classifier – (*scene_chunks*). Each *scene_chunk* is padded with *pad_time* to its left and right; see supplement for more details.

1. **Text:** we use the ASR output to extract the words spoken during each chunk in *scene_chunks*. Using the method described in **Text Extraction using ASR and text denoising** subsection, we extract the Medical and ROI caption for this chunk.

2. **Image:** we extract representative image(s) for every chunk/time-interval in *scene_chunks* as described in **Filtering for narrative-style medical videos** subsection above.

Finally, each chunk in *scene_chunks* is mapped to texts (both medical and ROI captions) and images. Next we map each medical image to one or more medical text. Using the time interval in which the image occurs, we extract its raw text from ASR and then correct and extract keywords using the Rake method, which we refer to as *raw_keywords*. We extract keywords from each medical text returned using the LLM, and we refer to these as *keywords*. Finally, if the *raw_keywords* occur before or slightly after a selected representative image, and overlap with the *keywords* in one of the Medical/ROI texts for that chunk, we map the image to the medical/ROI text. Example. *keywords*: *psammoma bodies*, match with *raw_keyword*: *psammoma bodies* within the ASR-corrected text '*Meningiomas typically have a meningothelial pattern with lobular-like arrangements and psammoma bodies*.' Refer to Figures in the supplement for a detailed explanation of the method and examples of aligned image and text.

### 3.2  QUILT-1M: Combining QUILT with other histopathology data sources

To create QUILT-1M, we expanded QUILT by adding other disparate histopathology image-text open-access sources: LAION, Twitter, and PubMed.

**PubMed Open Access Articles.** We searched the PubMed open-access from 2010-2022, extracting 59,371 histopathology image-text pairs, using our histopathology classifier and multi-plane figure cropping algorithm. The images are categorized into (1) images that are fully histopathology, (2) multi-plane images that contain histopathology sub-figures, and (3) histopathology sub-figures cropped from (1) and (2). See supplement for more details.

**Histopathology Image Retrieval from LAION.** The Large-scale Artificial Intelligence Open Network (LAION-5B) [52] curated over 5 billion pairs of images and text from across the Internet, including a substantial volume of histopathology-related data. We tapped into this resource by retrieving 22,682 image and text pairs. See supplement for more details.

**Twitter Data from OpenPath.** We utilized a list of tweets curated by Huang et al. [24], which totaled up to 55,000 unique tweets and made up $133,511$ unique image-text pairs. This exhibits a one-to-many relationship where many images were matched with multiple captions; this differentiated our work from the OpenPath approach. To maintain comparability, we followed their text pre-processing pipeline [24]. See supplement for more details.

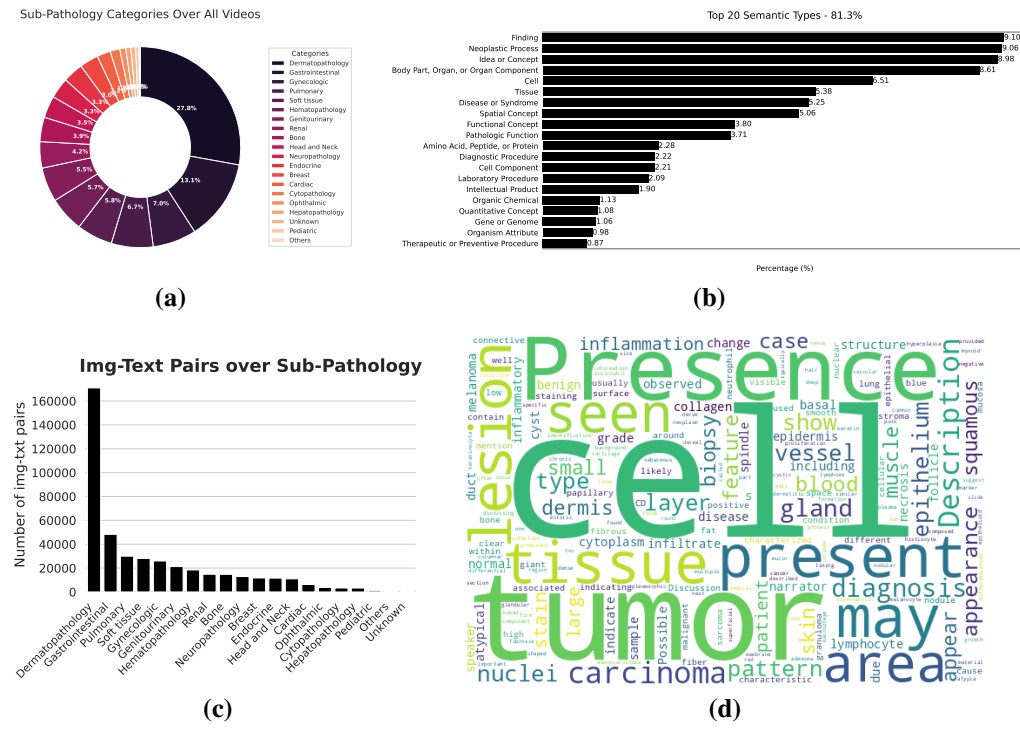

Figure 4: **(a)** Distribution of all videos over sub-pathology types. **(b)** Distribution of our entities across top 20 UMLS semantic types. **(c)** Number of image-text pairs within each sub-pathology type. **(d)** word cloud of all the text in QUILT.

## 3.3 Quality

To evaluate our pipeline's performance, we assess several aspects. First, we calculate the precision of our LLM's corrections by dividing the number of *conditioned* misspelled errors replaced (i.e., passed the UMLS check) by the total number of *conditioned* misspelled words found, yielding an average of 57.9%. We also determined the *unconditioned* precision of the LLM, similar to the previous step, and found it to be 13.8%. Therefore, we replace our detected incorrect words with the LLM's correction 57.9% of the time, and 13.8% of the time we replace the LLM's detected errors with its correction (see supplement for more details). To estimate the ASR model's transcription performance, we compute the total number of errors replaced (both conditioned and unconditioned) and divide it by the total number of words in each video, resulting in an average ASR error rate of 0.79%. To assess the LLM's sub-pathology classification, we manually annotated top-k ($k = 1, 2, 3$) sub-pathology types for 100 random videos from our dataset. The LLM's accuracy for top-3, top-2, and top-1 was 94.9%, 91.9%, and 86.8%, respectively. Also note that, by prompting the LLM to extract only medically relevant text, we further eliminate identifiable information, such as clinic addresses, from our dataset.

## 3.4 Final dataset statistics

We collected QUILT, from $4475$ narrative videos spanning over $1087$ hours with over $437K$ unique images with $802K$ associated text pairs. The mean length of the text captions is $22.76$ words, and $8.68$ words for ROI text, with an average of $1.74$ medical sentences per image (max=5.33, min=1.0). Our dataset spans a total of $1.469M$ UMLS entities from those mentioned in the text (with 28.5K unique). The images span varying microscopic magnification scales (0-10x, 10-20x, 20-40x), obtaining (280K,

75K, 107K) images from each scale respectively with an average height and width of 882 x 1468 pixels, as we leverage the max image resolution of videos. Figure 4 (a, c) plots our dataset's diversity across multiple histopathology sub-domains. This plot shows that the captions cover histopathology-relevant medical subtypes: findings, concepts, organs, neoplastic processes, cells, diseases, and a mix of laboratory and diagnostic procedures. Overall, across all 127 UMLS semantic types, our entities cover 76.2% of medically-related semantic types (e.g., findings, disease, or syndrome) and 23.75% non-medical (e.g., geographic area, governmental or regulatory activity).

# 4   QUILTNET: Experiments training with QUILT-1M

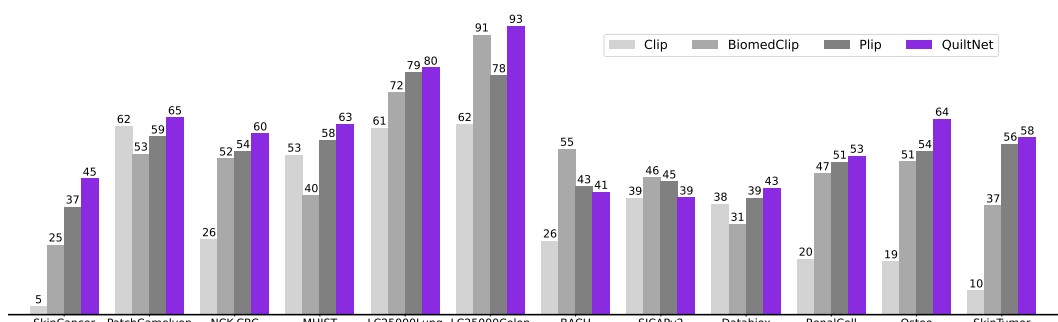

Figure 5: QUILTNET, outperforms out-of-domain CLIP baseline and state-of-the-art histopathology models across 12 zero-shot tasks, covering 8 different sub-pathologies (accuracy percentage provided).

We use the Contrastive Language-Image Pre-training (CLIP) objective [46] to pretrain QUILTNET using QUILT-1M. CLIP takes a batch of $N$ (image, text) pairs and optimizes a contrastive objective to create a joint embedding space. The optimization process involves concurrent training of both image and text encoders to increase the cosine similarity of embeddings from aligned pairs, while decreasing it for unaligned pairs. The objective is minimized via the InfoNCE loss, expressed as:

$$\mathcal{L} = -\frac{1}{2N}\left(\sum_{i=1}^{N}\log\frac{e^{\cos(I_i,\boldsymbol{T}_i)}}{\sum_{j=1}^{N}e^{\cos(I_i,\boldsymbol{T}_j)}} + \sum_{i=1}^{N}\log\frac{e^{\cos(\boldsymbol{I}_i,\boldsymbol{T}_i)}}{\sum_{j=1}^{N}e^{\cos(I_j,T_i)}}\right)$$

where $I_i$ and $T_i$ are the embeddings for the aligned $i$-th image and text, respectively. For the image encoder, we use both ViT-B/32 and ViT-B/16 architectures [16]. For the text encoder, we use GPT-2 [45] with a context length of 77, and PubmedBert [67]. We train QUILTNET by finetuning an OpenAI pre-trained CLIP model [46] on QUILT-1M to enhance its performance in histopathology. Once finetuned, we conduct experiments on two types of downstream tasks: image classification (zero-shot and linear probing) and cross-modal retrieval (zero-shot). We also compare the performance of fine-tuning a pre-trained CLIP model versus training it from scratch.

**Downstream histopathology datasets.** We evaluate the utility of QUILTNET on 13 downstream datasets: **PatchCamelyon** [57] contains histopathology scans of lymph node sections labeled for metastatic tissue presence as a binary label. **NCT-CRC-HE-100K** [31] consists of colorectal cancer images and is categorized into cancer and normal tissue. For **SICAPv2** [53] the images are labeled as non-cancerous, Grade 3-5. **Databiox** [8] consists of invasive ductal carcinoma cases of Grades I-III. **BACH** [4] consists of breast tissues labeled as normal, benign, in-situ, and invasive carcinoma. **Osteo** [5] is a set of tissue patches representing the heterogeneity of osteosarcoma. **RenalCell** [10] contains tissue images of clear-cell renal cell carcinoma annotated into five tissue texture types. **SkinCancer** [34] consists of tissue patches from skin biopsies of 12 anatomical compartments and 4 neoplasms that make up the **SkinTumor** Subset. **MHIST** [60] contains tissue patches from Formalin-Fixed Paraffin-Embedded WSIs of colorectal polyps. **LC25000** [9], which we divide into **LC25000 (Lung)** and **LC25000 (Colon)**, contains tissue of lung and colon adenocarcinomas. For more details see supplemental material.

**Results using zero-shot learning**. Given the vast diversity of cancer sub-types in histopathology, it is critical that a model maintains comprehensive understanding without requiring specific data

Table 1: **Linear probing**. Classification results, denoted as accuracy % (standard deviation). Camelyon denotes the PatchCamelyon dataset. Supervised results are from each dataset's SOTA models.

| Dataset | %shot | ViT-B/32 | | | ViT-B/16 | | | |
|---|---|---|---|---|---|---|---|---|
| | | CLIP | PLIP | QUILTNET | CLIP | QUILTNET | BiomedCLIP | QUILTNET |
| Supervised(%) | | GPT/77 | GPT/77 | GPT/77 | GPT/77 | GPT/77 | PMB/256 | PMB/256 |
| NCT-CRC [31] | 1 | 91.0 (0.10) | 93.75 (0.09) | **94.64**(0.22) | 90.96 (0.10) | 93.36 (0.23) | 92.14 (0.12) | **93.55** (0.25) |
| | 10 | 92.02 (1.30) | 93.83 (0.06) | **95.30** (0.03) | 92.58 (0.12) | **93.85** (0.04) | 92.90 (0.07) | 93.72 (0.08) |
| (94.0) | 100 | 91.83 (0.01) | 94.16 (0.01) | **95.22** (0.01) | 92.26 (0.09) | **93.76** (0.02) | 92.97 (0.05) | 93.60 (0.01) |
| Camelyon [57] | 1 | 80.38 (0.16) | 87.26 (0.23) | **87.62** (0.35) | 80.28 (0.20) | **84.78** (0.14) | 83.63 (0.44) | 83.48 (0.18) |
| | 10 | 82.67 (0.19) | 87.48 (0.08) | **87.55** (0.03) | 82.20 (0.04) | **86.77** (0.09) | 84.18 (0.15) | 84.42 (0.10) |
| (97.5) | 100 | 82.80 (0.01) | 87.34 (0.01) | **87.48** (0.01) | 82.55 (0.02) | **86.81** (0.04) | 84.23 (0.01) | 84.44 (0.02) |
| SkinCancer [34] | 1 | 84.27 (0.22) | **91.07** (0.25) | 90.93 (0.25) | 85.62 (0.16) | **88.29** (0.15) | 87.53 (0.21) | 88.06 (0.20) |
| | 10 | 89.0 (0.02) | **93.39** (0.05) | 92.99 (0.02) | 90.28 (0.01) | **91.20** (0.0) | 89.23 (0.03) | 90.03 (0.02) |
| (93.3) | 100 | 89.02 (0.02) | **93.29** (0.01) | 93.03 (0.02) | 90.29 (0.03) | **91.20** (0.0) | 89.16 (0.02) | 89.91 (0.01) |
| SICAPv2 [53] | 1 | 52.45 (2.41) | 65.76 (2.65) | **69.92** (1.02) | 56.01 (0.66) | 66.86 (1.16) | **69.43** (1.03) | 68.49 (1.06) |
| | 10 | 62.24 (0.65) | 69.23 (0.43) | **74.14** (0.38) | 63.70 (0.69) | 72.37 (0.65) | 71.61 (0.31) | **72.48** (0.42) |
| (67.0) | 100 | 65.75 (0.16) | 73.0 (0.14) | **75.48** (0.12) | 68.74 (0.10) | 74.14 (0.16) | 74.57 (0.04) | **74.60** (0.17) |

for retraining. Thus, we evaluate our model's zero-shot performance against three state-of-the-art models: CLIP, BiomedCLIP, and PLIP. Our model demonstrates superior performance, as illustrated in Figure 5, where it outperforms the other models in all but two datasets, in which BiomedCLIP performs marginally better. See supplemental material for UMap visualizations and cross-modal attention visualization comparison. The prompts used for these evaluations are presented in the supplemental material. To ensure a fair comparison with BiomedCLIP, which uses a ViT-B/16 and PMB/256 (pre-trained with [67]), we trained three different variants of our model. For detailed insights into the results, please refer to supplemental material.

**Results using linear probing.** We assess the few-shot and full-shot performance of our model by conducting linear probing with 1%, 10%, and 100% of the training data, sampled with three different seeds; we report the average accuracy and their standard deviation in Table 1. We deploy our evaluation across four distinct datasets, specifically those with dedicated training and testing sets among our external datasets. Remarkably, our model, utilizing the ViT-B/32 architecture with GPT/77, outperforms its counterparts, BiomedCLIP, PLIP, and CLIP, in most datasets. On the NCT-CRC and SICAPv2 datasets, our model surpasses even the fully supervised performance using only 1% of the labels. Also, note that for some results 10% does better than 100%; this is because we are sampling from each class equally, and thus the 10% subset contains a more balanced training set than 100%, for datasets that are very imbalanced, resulting in sub-optimal performance at 100%.

**Results using cross-modal retrieval.** In our study, we evaluate cross-modal retrieval efficacy by examining both zero-shot text-to-image and image-to-text retrieval capabilities. We accomplish this by identifying the nearest neighbors for each modality and then determining whether the corresponding pair is within the top $N$ nearest neighbors, where $N \in \{1, 50, 200\}$. Our experiments are conducted on two datasets: our holdout dataset from QUILT-1M and the ARCH dataset. Results are in Table 2.

## 5 Discussion

**Limitations.** Despite the promising results, QUILT was curated using several handcrafted algorithms and LLMs. Such curation methods, while effective, introduce their own biases and errors. For instance, our histopathology classifier had occasional false positives ($\approx 5\%$) confirmed by human evaluation. Occasionally, ASR can misinterpret a medical term and transcribe it as a different existing term, such as transcribing 'serous carcinoma' as 'serious carcinoma'. Unfortunately, such errors are not rectifiable using our current pipeline (see supplement for more details). While not directly a limitation of our dataset, training a CLIP model trained from scratch underperformed compared

Table 2: Cross-modal retrieval results on the Quilt-1M holdout set and ARCH dataset. In each cell, the results are displayed in the format (%/%), with Quilt-1M holdout results on the left and ARCH results on the right. The best-performing results are highlighted in bold text.

| model | config | Text-to-Image (%) | | | Image-to-Text (%) | | |
|---|---|---|---|---|---|---|---|
| | | R@1 | R@50 | R@200 | R@1 | R@50 | R@200 |
| CLIP | ViT-B/32\|GPT/77 | 0.49/0.07 | 4.73/2.42 | 10.15/7.21 | 0.39/0.05 | 3.99/2.52 | 8.80/7.22 |
| PLIP | ViT-B/32\|GPT/77 | 1.05/0.56 | 10.79/13.10 | 21.80/29.85 | 0.87/0.74 | 11.04/13.75 | 21.63/29.46 |
| QuiltNet | ViT-B/32\|GPT/77 | **1.17/1.41** | **16.31/19.87** | **31.99/39.13** | **1.24/1.35** | **14.89/19.20** | **28.97/38.57** |
| CLIP | ViT-B/16\|GPT/77 | 0.83/0.09 | 5.63/2.73 | 11.26/8.72 | 0.66/0.13 | 5.02/3.09 | 10.82/9.04 |
| QuiltNet | ViT-B/16\|GPT/77 | 2.42/1.29 | 22.38/20.30 | 41.05/40.89 | 2.00/1.01 | 21.66/16.18 | 39.29/34.15 |
| BiomedCLIP | ViT-B/16(224)\|PMB/256 | 4.34/**8.89** | 14.99/53.24 | 25.62/71.43 | 3.88/**9.97** | 13.93/52.13 | 23.53/68.47 |
| QuiltNet | ViT-B/16(224)\|PMB/256 | **6.20**/8.77 | **30.28/55.14** | **50.60/77.64** | **6.27**/9.85 | **31.06/53.06** | **50.86/73.43** |

to fine-tuning a pre-trained CLIP (see supplement for more details). This suggests that a million image-text pairs may still not be sufficient. Future works may explore other self-supervised objectives.

**Data Collection and Societal Biases** Aligning in strategies with [65], we release Quilt derived from public videos, taking structured steps to limit privacy and consent harms (see supplement for more details). Complying with YouTube's privacy policy, we only provide video IDs, allowing users to opt-out of our dataset. Researchers can employ our pipeline to create Quilt. Regarding societal biases, a significant portion of our narrators originate from western institutions, a situation that is further amplified by our focus on English-only videos. Consequently, QuiltNet may exhibit inherent biases, potentially performing better on data associated with these demographics, while possibly underperforming when applied to other cultural or linguistic groups.

**Conclusion.** We introduced Quilt-1M, the largest open-sourced histopathology dataset to date. Empirical results validate that pre-training using Quilt is valuable, outperforming larger state-of-the-art models like BiomedCLIP across various sub-pathology types and tasks including zero-shot, few-shot, full-shot, and cross-modal retrieval. We established a new state-of-the-art in zero-shot, linear probing, and cross-modal retrieval tasks in the field of Histopathology.

## Acknowledgments

Research reported in this study was supported by the National Cancer Institute under Awards No. R01 CA15130, R01 CA225585, and R01 CA201376 and the Office of the Assistant Secretary of Defense for Health Affairs through the Melanoma Research Program under Awards No. W81XWH-20-1-0797 and W81XWH-20-1-0798. Opinions, conclusions, and recommendations are those of the authors.

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

# A  Data curation models, algorithms and parsing pipelines

## A.1  Curating QUILT: an Overview

Creating a densely annotated vision-language dataset from videos is a significant undertaking, as it involves various handcrafted algorithms and machine learning models. In the following sections, we present more detailed information about the challenges of the data curation pipeline and algorithms used to address these challenges. To download QUILT-1M and its metadata and access the code to recreate the dataset and trained models, refer to our website.

**Collecting representative channels and videos.**  The first challenge lies in obtaining relevant histopathology videos. We used a set of keywords (obtained from online histopathology glossaries [7]) to search for videos, resulting in $\approx$ 65K potential matches. Figure 6 shows the word cloud of all keywords used for searching YouTube. However, filtering histopathology content based on thumbnail and title yields many false positives, often including general pathology videos. To address this, we process the frames of lower-resolution versions of each video to differentiate between histopathology and pathology content, narrowing the selection to $\approx$ 9K videos.

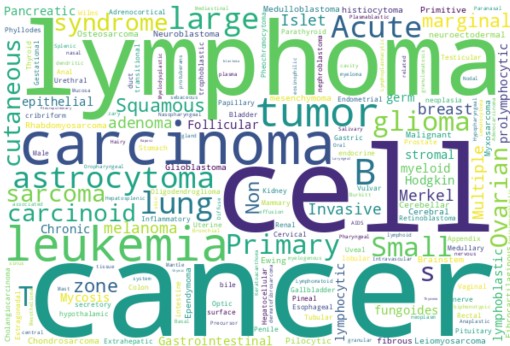

Figure 6: Word cloud of all keywords used for searching YouTube

**Filtering for narrative-style medical videos.**  Among the $\approx$ 9K videos, we sought videos with a "narrative style" where narrators freely explain whole slide images and streaks of similar frames occur, indicating an educational performance. To identify such content, we used a model that analyzed randomly sampled frames to determine if they maintained a consistent style over time. This process resulted in the selection of $\approx$ 4K videos. Non-voiced videos are also filtered by using inaSpeechSegmenter [17] where the video endpoint does not provide the video language or transcript. To identify the audio language of a video, we first check YouTube's API. If the information is unavailable through the API, we use OpenAI's Whisper model [47] on the first minute of audio from the video.

To identify videos containing medical content, we employ a keyframe extraction process with a specific threshold to determine the minimum visual change required to trigger keyframes. For a new video, the thresholds for keyframe extraction are determined by linearly interpolating between the lowest threshold, $0.008$ (5-minute video) and the highest $0.25$ (200-minute video). Following the keyframe extraction process, we utilize a histopathology image classifier to identify histopathology content within the extracted keyframes. See A.3 for more details. To identify narrative-style videos, we randomly select a $min($num_of_histo_scene_frames$, 20)$ keyframes from a video and utilize a pre-trained CLIP [8] (ViT-B-32) model to embed and compute a cosine similarity on the next three keyframes. If all three have similarity scores $\geq$ a threshold of $0.9$, we count the video as a narrative streak.

**Text extraction using ASR and text denoising.**  Another challenge involves automatic speech recognition (ASR), as YouTube captions are often inadequate for medical vocabulary. To address this issue, we employed the Large-V2 open-source Whisper model [47] for speech-to-text conversion. However, general-purpose ASR models like Whisper can misinterpret medical terms, particularly

---

[7]https://lab-ally.com/histopathology-resources/histopathology-glossary

[8]https://huggingface.co/sentence-transformers/clip-ViT-B-32

Table 3: Salvagable and Non-salvagable cases for ASR correction using an LLM.

| Error due to | Raw output | Salvagable (beacause LLM can rephrase and/or extract contextually similar correction) | Non-salvagable (because the error losses all possible medical context and can lead to wrong entries) |
|---|---|---|---|
| Unfinetuned ASR | ...look like the cranialomas I would expect in HP. They actually look more sarcoidal to me. The reason I say that is they, there's a kind of positive of inflammatory cells associated with them. They're really tight and well-formed. They're very easy to see a low power. And so HP is in the differential hypersensium nitose, but I would be more worried about sarcoidosis. | differential hypersensium nitose: hypersensitivity pneumonitis, cranialomas: granulomas | positive: paucity |
| LLM | high-larbidia-stinal lymphadenocathy

lymphin-giatic pattern distribution | returns hilar lymphadenopathy instead of a more appropriate hilar mediastinal lymphadenopathy | returns lymphatic pattern distribution instead of a more appropriate lymphangitic pattern distribution |
| Incomplete UMLS checker | ...picnotic | - | LLM correctly returns pyknotic however, UMLS(2020) does not have the word pyknotic if fails to pass the UMLS check. |

when the speaker's voice is choppy or accented. There are no straightforward trivial solutions due to: **1)** the absence of openly available medical ASR models or data for fine-tuning in the medical domain; **2)** the inadequacy of medical named entity recognition models in detecting transcription errors, because these models are typically trained on correctly spelled words; **3)** the ineffectiveness of methods like semantically searching over a medical glossary, such as UMLS, which only prove effective when the erroneous text has significant similarity to the correct terms; and **4)** the inability of simpler methods like finding the longest common substring, which might work in finding a match in the glossary/ontology for replacement, but cannot identify the wrong words/phrases in the first place. To rectify ASR errors, we employed UMLS (a knowledge database) and a LLM (GPT-3.5). This, however, introduces a new challenge of identifying incorrectly transcribed words and determining which words were mistakenly "corrected" and correctly formatted by the LLM after error correction and resolving unintended parsing errors [1]. See Figure 3 in the main paper for LLM prompt examples of ASR correction and medical and ROI text extraction from the corrected ASR text. Refer to Table 3 for error examples of ASR correction using the LLM.

**Image frame extraction and denoising.** The image processing aspect of this task adds to its complexity, as it requires static frame detection, quality control for frames, and histology magnification classification. Each model utilized it these steps introduces its own biases and errors. We extract time-intervals (*chunks*) from each video from which we extract representative image(s). For each of the extracted *chunks* $(t_n, t_{n+1})$, the static chunk detection algorithm 1 is used to extract sub-time-intervals with static frames within the chunk. If found, we save the median (in pixel space to prevent blurry outputs) of the stable frames, else (i.e no stable duration of frames) we leverage the structural similarity index (SSIM) method on histopathology key-frames to find the most dissimilar histopathology image to make up the representative images for the chunk, essentially de-duplicating the frames. Figure 7 demonstrates this process.

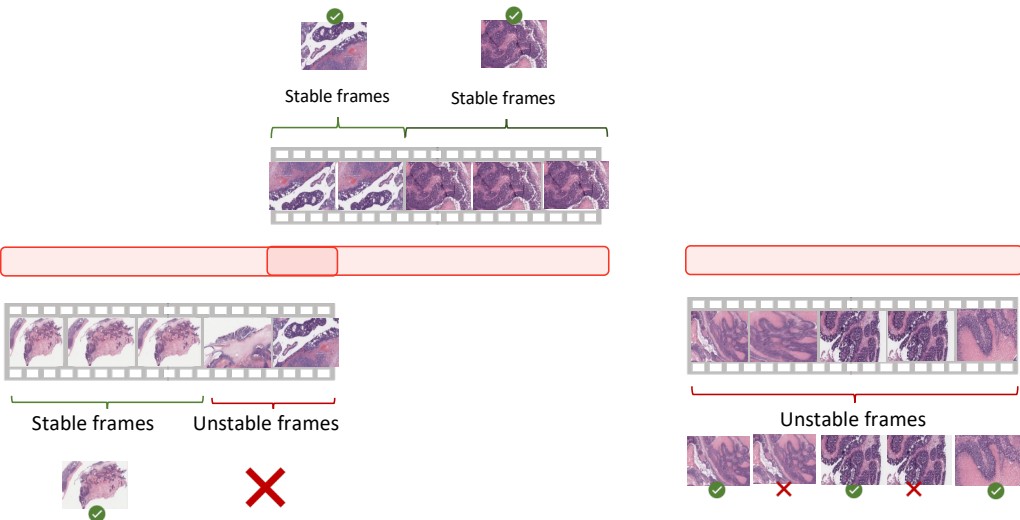

Figure 7: Representative Frame Identification. If a Stable frame is found by Algorithm 1 within the candidate regions, we use it as the representative frame. If not, we use the most dissimilar frames among unstable frames.

---

**Algorithm 1** Static Video Chunk Detection Algorithm

---

1: **procedure** DETECTSTATICFRAMES(video, starttime, endtime)
2:     video = video[starttime:endtime]
3:     $fixedFrames \leftarrow \emptyset$
4:     $SSIMValidatedFrames \leftarrow \emptyset$
5:     $prevFrame \leftarrow$ first frame in $video$
6:     **for** $frame \in$ rest of frames in $video$ **do**
7:         $absDiff \leftarrow$ absolute difference between $frame$ and $prevFrame$
8:         $absDiffThresh \leftarrow$ apply adaptive thresholding using a Gaussian filter to $absDiff$
9:         $meanVal \leftarrow$ mean value of $absDiffThresh$
10:         **if** $meanVal < 10$ **then**
11:             $fixedFrames \leftarrow fixedFrames \cup frame$
12:         **else**
13:             **if** length of $fixedFrames \geq$ minimum duration **then**
14:                 $subclip \leftarrow$ extract sub-clip of frames with constant background from $fixedFrames$
15:                 **for** $patch \in$ randomly selected patches in each frame of $subclip$ **do**
16:                     $SSIMVal \leftarrow$ calculate SSIM of $patch$
17:                     **if** $SSIMVal >$ threshold **then**
18:                         $SSIMValidatedFrames \leftarrow SSIMValidatedFrames \cup frame$
19:                     **end if**
20:                 **end for**
21:             **end if**
22:             $fixedFrames \leftarrow \emptyset$
23:         **end if**
24:         $prevFrame \leftarrow frame$
25:     **end for**
26:     $staticTimestamps \leftarrow$ extract start and end times from SSIMValidatedFrames
27:     **return** $staticTimestamps$
28: **end procedure**

---

**Aligning both modalities.** The alignment of the images with their corresponding text requires the implementation of unique algorithms. These algorithms are designed to reduce duplicate content and ensure accurate mappings between image and text. See Figures 8 and 9 and Table 4 for a

a demonstration of image-text alignment process. See Figure 10 for sample images and their corresponding medical and ROI texts and the sub-pathology classification provided by the LLM.

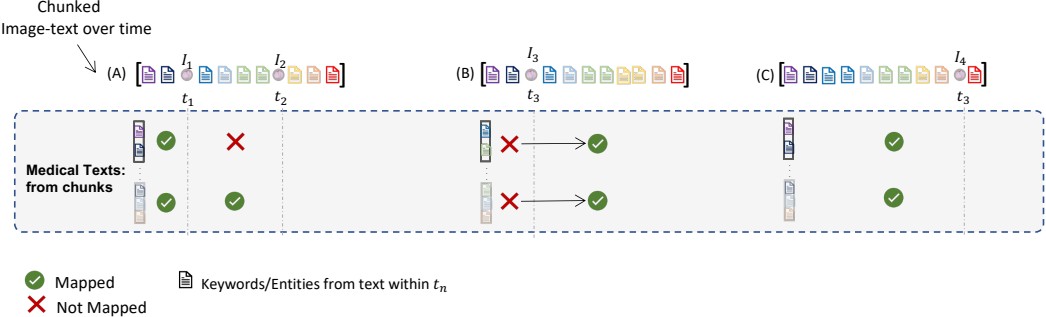

Figure 8: **Overview of use of timing and keywords for Alignment** Images within a video chunk, i.e {A, B, C}, $I_n$ at $t_n$ are aligned with medical texts extracted within the same chunk. The *raw_keywords* within each example chunk is colour coded to illustrate matches with *keywords* extracted from the medical texts and only matching keywords allow for the pairing of texts containing said *keywords* to image frames with frame-times around *raw_keywords* times.

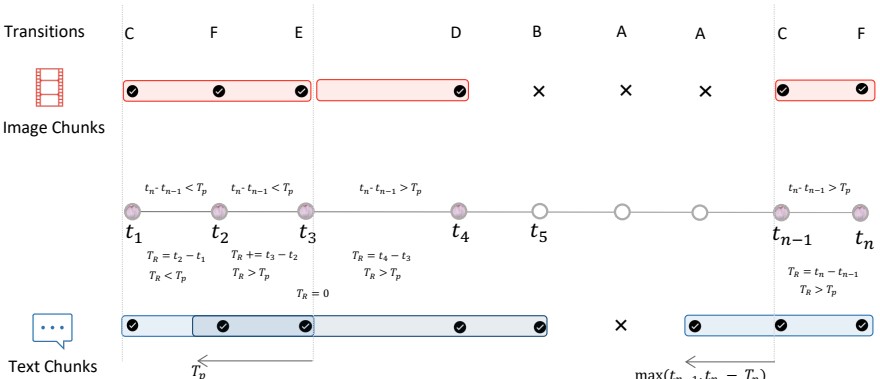

Figure 9: **Video Chunking algorithm illustrate.** With each transition tag explained in Table 4, we leverage predicted histopathology frames at times / $t_1, \cdots t_n$/ to segment videos into chunks. Chunks at are minimum are $T_P$ in duration, this value is estimated based on the word-per-second of the video with a minimum of 20 words being captured per chunk. Images within a chunk, unlike texts, are not overlapping with other chunks . Text overlap is done to provide needed context for LLM text correction and extraction.

## A.2 Other data sources

### A.2.1 PubMed Open Access Articles

We searched the PubMed open-access from $2010 - 2022$ with keywords (pathology, histopathology, whole-slide image, H&E, and $148$ keywords from a histopathology glossary[9]). We utilized Entrez [10] to retrieved the top 10,000 most relevant articles for each keyword. This query yielded 109,518 unique articles with PMCIDs. We extracted $162,307$ images and their corresponding captions. Using our histopathology classifier and cropping multi-plane figures as described in A.4, we extracted $59,371$ histopathology image and caption pairs with an average caption length of $54.02$ tokens. Figure 11 demonstrates the pipeline of collecting data from PubMed.

---

[9]https://lab-ally.com/histopathology-resources/histopathology-glossary
[10]http://www.ncbi.nlm.nih.gov/Entrez

Table 4: **All 6 (six) transition states for chunking narrative style videos.** $p(H)_{t_n}$ is the binary histo image classifier prediction at the current frame's time $t_n$ and $p(H)_{t_{n-1}}$ is the prediction at next frame's time $t_{n-1}$, where $T_R$ is the cumulative running time and $T_P$ is the estimated minimum chunk time for the video, determined by the words per second of the video. Text and image chunks are implemented as an ordered list of time intervals and image indexes.

| $P(H)@t_n$ | $P(H)@t_{n-1}$ | $t_n - t_{n-1} > T_p$ | $T_r > T_p$ | Text chunk | Image chunk | Tag |
|---|---|---|---|---|---|---|
| 0 | 0 | – | – | – | – | A |
| 0 | 1 | – | – | $end = t_n$; append$(s, e)$; reset | append index to chunk state, if state is empty append prior index; reset state | B |
| 1 | 0 | – | – | $start = \max(t_{n-1}, t_n - T_p)$ | append index to chunk state | C |
| 1 | 1 | 1 | – | $end = t_n$; append$(s, e)$; reset state; $start = t_n - T_p$ | append index to chunk state; reset state | D |
| 1 | 1 | 0 | 1 | $end = t_n$; append$(s, e)$; reset state; $start = t_n - T_p$ | append index to chunk state; reset state | E |
| 1 | 1 | 0 | 0 | – | append index to chunk state | F |

### A.2.2 Histopathology Image Retrieval from LAION

The Large-scale Artificial Intelligence Open Network (LAION-5B) [52] curated over 5 billion pairs of images and text from across the Internet, including a substantial volume of histopathology-related data. We tapped into this resource by retrieving the 3000 most similar LAION samples for each of the $1,000$ pairs of images and text sampled from PubMed and QUILT, using a CLIP model pre-trained on the LAION data. The retrieval process utilized both image and text embeddings, with cosine similarity serving as the distance metric. Subsequently, we eliminated the duplicate images and removed all non-English pairs from the remaining pairs using LangDetect[11]. Consequently, the process yielded $22,682$ image and text pairs.

### A.2.3 Twitter Data from OpenPath

We utilized a list of tweets curated by Huang et al. [24] which totaled up to $55,000$ unique tweets and $133,511$ unique image-text pairs. This exhibits a one-to-many relationship that leans towards the image side, differentiating our work from the OpenPath approach, where we had one image matching with multiple captions (as in the case of MS-COCO captions). In order to maintain comparability with OpenPath, we followed their text pre-processing pipeline given in [24].

### A.3 Histopathology and Magnification classifier

We use an ensemble of three histopathology image classifiers. To ensure robustness, our ensemble approach consists of two small Conv-NeXt models [38] and one linear classifier fine-tuned with DINO features [11]. This combination is necessary due to the homogenous appearance of histopathology images and the risk of false positives from similar pinkish-purple images. One Conv-NeXt model is trained in detecting non-H&E Immunohistochemistry (IHC) stained tissue images, while the other models are trained to handle all IHC stains and tissue types. The training data includes eight subgroups of the TCGA WSI dataset and a mix of general-domain images, PowerPoint (slide) images, and scientific figure datasets. See Table 5 for details of these datasets.

For the magnification classifier, we finetune a pretrained ConvNeXt-Tiny model [38], with standard preset hyperparameters for a few epochs and select the best performing model on the validation set. To generate a training set for the magnification model, TCGA subsets were segmented into patches using a method similar to [64]. These patches were generated at various magnifications, which were then categorized into three labels: 0:{1.25x, 2.5x, 5x, 10x}, 1:{20x}, 2:{40x}. The TCGA subsets were chosen to ensure a diverse representation of tissue morphologies and cancer types, thereby ensuring robust and comprehensive model training. The model was also trained on cytopathology microscopy images and various IHC stains beyond H&E to enhance the model's generalizability

---

[11]https://github.com/fedelopez77/langdetect

| Image | Medical TEXT | ROI Text | Sub-pathology Classification |
|---|---|---|---|
|  | ['There are clusters of cells with micro-follicular formations.','Nuclear pseudo-inclusions, oval nuclei, nuclear grooves, and small nucleoli are present in some cells.'] | ['clusters of cells', 'micro-follicular formations', 'nuclear pseudo-inclusions', 'oval nuclei', 'nuclear grooves', 'small nucleoli'] | ['Endocrine', 'Cytopathology', 'Head and Neck'] |
|  | ['Cluster of macrophages and T cells is characteristic of acute rheumatic fever.', 'Aschoff body is a characteristic feature of acute rheumatic fever.', 'Macrophages with elongated chromatin are called Anitchkow cells and are commonly seen in Aschoff bodies.', 'Pancarditis with Aschoff bodies is present.'] | ['Cluster of macrophages and T cells', 'Aschoff body', 'Macrophages with elongated chromatin', 'Anitchkow cells', 'Pancarditis'] | ['Cardiac', 'Hematopathology', 'Endocrine'] |
|  | ['An 80-year-old man has a scar-like plaque on the scalp that has been called malignant on a biopsy.', 'The tissue affected by the plaque extends from the epidermis to the galea aponeurotica, near the periosteum of the skull.', 'The skin, dermis, and subcutis are all affected by the process.'] | ['scar-like plaque on the scalp', 'malignant on a biopsy', 'skin, dermis, and subcutis affected by the process'] | ['Dermatopathology', 'Soft tissue', 'Hematopathology'] |
|  | ['Inflammatory cells surrounding cartilage can indicate acute chondritis, with neutrophils being the principal cell type.', 'Chronic chondritis may be diagnosed if lymphocytes are the predominant inflammatory cell type.'] | ['cartilage', 'inflammatory cells'] | ['Hematopathology', 'Bone', 'Dermatopathology'] |
|  | ['Large histiocytes with abundant cytoplasm identified as Rosai-Dorfman histiocytes.', 'S100 stain showed perivascular cuffing.', 'Initial diagnosis of inflammatory pseudotumor of the orbit.', 'Rosai-Dorfman disease may burn out and leave behind fibrotic pockets.'] | ['Large histiocytes', 'perivascular cuffing', 'fibrotic pockets'] | ['Dermatopathology', 'Soft tissue', 'Hematopathology'] |
|  | ['Epidermal acanthosis and papillomatosis resembling a wart or seborrheic keratosis.', 'Presence of large sebaceous glands that drain directly through their duct out to the skin surface, which is abnormal.', 'Presence of a demodex mite.'] | ['Epidermal acanthosis and papillomatosis', 'large sebaceous glands', 'demodex mite'] | ['Dermatopathology', 'Soft tissue', 'Hematopathology'] |
|  | ['Histological description of glandular tissue with little atypia but located in a place where it does not belong can be a helpful criteria to discern the presence of malignancy.', 'Glands located on the periphery and infiltrating into adventitia and peripancreatic tissue may be malignant.'] | ['glandular tissue', 'pancreas',] | ['Gastrointestinal', 'Pancreatic', 'Hematopathology'] |

Figure 10: A collection of sample images from our dataset, accompanied by corresponding medical text, ROI text, and the top three sub-pathology classifications derived from the ASR text using the LLM.

across different conditions. Only the ACROBAT and TCGA datasubsets are preprocessed to divide the WSIs into patches at various scales.

## A.4 Support Models, Ontology Databases and Algorithms

This section describes the support models, ontology databases and handcrafted algorithms utilized within our pipeline for both searching and parsing our data.

**Ontology databases.** We employ various ontologies, both specific to histopathology and general ones. Among them are OCHV [2], FMA [42], BCGO [12], NCIT [19], MPATH [51], HPATH [62], and CMPO [29]. These ontologies serve a dual purpose. First, we used histopathology-specific ontologies (HPATH, MPATH, BCGO, and CMPO) to provide words/phrases to condition the LLM, enabling it to identify incorrect words. Second, all ontologies, in conjunction with UMLS, are used to obtain terms or phrases for validating the output of the LLM.

---
[12]https://bioportal.bioontology.org/ontologies/BCGO

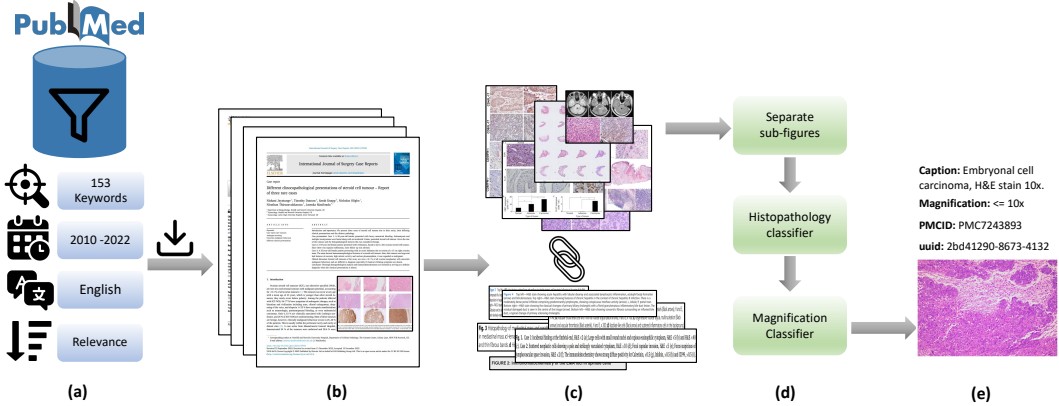

Figure 11: (a) Search PubMed open access database, filter based on keywords, date, language and sort by relevance. (b) Download paper and media for each search result. (c) Extract and pair figures and captions. (d) Separate multi-plane figures, find histopathology images and their magnification. (e) Final result.

Table 5: Datasets used to train the histopathology image classifier. [μm per pixel - MPP]

| Data Source | Subset | #WSI | #pathces | Train-Test | Magnification | Image-size |
|---|---|---|---|---|---|---|
| TCGA (H&E Stain) | GBM | 19 | 169,431 | 84715-16943 | 89,022 - 40x | 384 x 384 |
| | LUSC | 20 | | | | |
| | LIHC | 20 | | | 57,671 - 20x | |
| | SARC | 23 | | | | |
| | KIRC | 16 | | | 16,660 - 10x | |
| | KICH | 4 | | | 4,748 - 5x | |
| | BRCA | 17 | | | 1,465 - 2.5x | |
| | SKCM | 19 | | | 466 - 1.25x | |
| ACROBAT Weitz et al. [61] | H&E KI67 ER , PGR, HER2 | 99 | 50589 | 28105-22484 | (10x, 5x, 2.5x) | 384 × 384 |
| BCI Liu et al. [36] | - | - | 4,870 | | 20x (0.46 MPP) | 1024 × 1024 |
| CCESD Liu et al. [35] | - | - | 686 | | 100x/400x | 2048 × 1536 |
| Smear Hussain et al. [25] | - | - | 963 | | 400x | 2048 × 1536 |
| Celeb Liu et al. [37] | - | - | 202,599 | 8,103-1,944 | - | - |
| Places Zhou et al. [69] | - | - | 36,550 | 2,109-1,372 | - | - |
| AI2D Kembhavi et al. [32] | - | - | 4,903 | 0.7-0.3% | - | - |
| DocFig Jobin et al. [27] | - | - | 33,004 | 0.8-0.2% | - | - |
| SciFig-pilot Karishma [30] | - | - | 1,671 | 0.8-0.2% | - | - |
| SlideImages Morris et al. [40] | - | - | 8,217 | 0.8-0.2% | - | - |
| TextVQA Singh et al. [54] | - | - | 28,472 | 0.8-0.2% | - | - |
| SlideShare-1M Araujo et al. [3] | - | - | 49,801 | 0.8-0.2% | - | - |

**Sub-pathology types.** The list of all 18 sub-pathology types used to prompt LLM on the text classification task are: ***Bone, Cardiac, Cyto, Dermato, Endocrine, Gastrointestinal, Genitourinary, Gynecologic, Head and Neck, Hemato, Neuro, Ophthalmic, Pediatric, Pulmonary, Renal, Soft tissue, and Breast Histopathology.*** Figure 12 provides the LLM prompt to retrieve the top three sub-pathology classification based on a given text.

**Pre-processing multi-plane figures.** Many figures in academic papers are multi-plane, which means a number of sub-figures (Charts, graphs, histopathology and non-histopathology sub-figures) are placed next to each other to make a larger figure. We extracted individual images from multi-plane figures to create multiple instance bags. To locate boundaries and white gaps between sub-figures, we utilized Sobel filters. Binary thresholding was then used to find the contours surrounding the sub-figures. We employ image size and image ratio thresholds to remove undesirable sub-figures and our histopathology classifier to maintain just histopathology sub-figures. We supply the histological sub-figures individually for this type of figure by appending "_[0-9]+" to the end of the multi-plane figure id. If a figure is divided into more than 5 sub-figures, we preserve the original image to ensure that the resolution of these sub-figures remains reasonable. Figure 13 shows an overview of this pre-processing step in different scenarios of successful and unsuccessful crops.

User Prompt:
Imagine you are a text classifier. Classify the given text into one of the following surgical pathology types namely: Bone, Cardiac, Cytopathology, Dermatopathology, Endocrine, Gastrointestinal, Genitourinary, Gynecologic, Head and Neck, Hematopathology, Neuropathology, Ophthalmic, Pediatric, Pulmonary, Renal, Soft tissue, Breast pathology. Output only the top 3 pathology types in an ordered python list

Few-shot examples:
"Radicular cyst arises within the periodontal ligament space, particularly the periapex from the epithelial cell of malassez. These radicular cysts are caused by inflammation following the death of the pulp extending into the periapical radix. Radicular cysts caused by inflammation are always associated with a non vital tooth."

"['Soft tissue', 'Dermatopathology', 'Hematopathology']"

INPUT:
"There is a lesion with slight thickening of the muscularis mucosa and submucosa. There is a subtle change in the lamina propria that doesn't look quite like normal stromal cells. Description of slight thickening of the muscularis mucosa and submucosa with subtle changes in the lamina propria. Highlighted field shows the changes more dramatically. Abnormal cells in the lamina propria that appear pink and spindly."

OUTPUT: "['Gastrointestinal', 'Soft tissue', 'Hematopathology']"

Figure 12: Prompting LLM with few-shot examples to extract the top three sub-pathology classification of a given text.

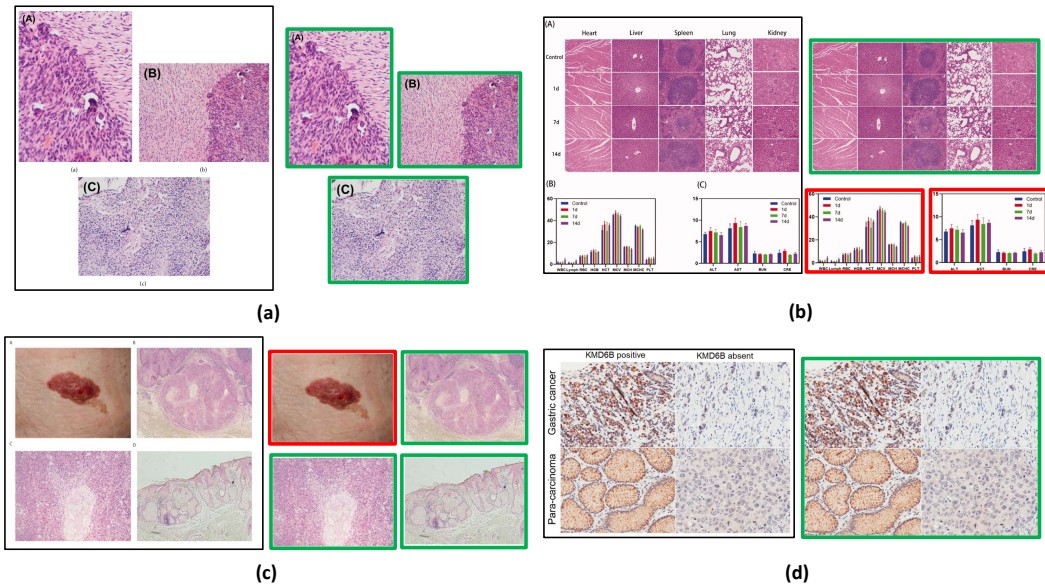

(a)  (b)

(c)  (d)

Figure 13: (a), (b), and (c) successfully cropped sub-figures where histopathology images (green box) are kept and non-histopathology (red box) images are removed. (b) histopathology crops are kept as not separated because the individual crops don't meet the size threshold so the original figure is kept. (d) Unsuccessful crop due to minimal gap between sub-figures. Original image is stored.

## A.5 Privacy preserving steps

In order to ensure privacy while handling the dataset, several steps were taken to protect sensitive information. These steps include:

- Reduction of Signal to Noise using a LLM: To protect the privacy of the dataset, a LLM was utilized to reduce the signal-to-noise ratio. By applying the LLM, irrelevant or sensitive information was masked or removed.

- Exclusion of Videos Not Fully in Narrative Style: Videos that did not adhere to a fully narrative style were intentionally left out of the dataset. This step was taken to minimize the risk of including any potentially private or sensitive content that could compromise individuals' privacy.

- Release of Video IDs and Reconstruction Code: Instead of directly releasing the complete dataset, only video IDs from YouTube were made public. Additionally, the code is provided to enable researchers to recreate the dataset.

- Collection from Diverse Channels: Data collection was performed from a wide range of sources, including both large and small channels. This approach aimed to decrease the risk of overfitting to specific channel types, thereby mitigating privacy concerns associated with potential biases linked to particular channels.

