# 🏳️ Quilt-1M: One Million Image-Text Pairs for Histopathology

**Wisdom O. Ikezogwo**[*]🏳️    **Mehmet S. Seyfioglu** 🏳️    **Fatemeh Ghezloo** 🏳️
**Dylan Geva**    **Fatwir S. Mohammed**    **Pavan K. Anand**
**Ranjay Krishna**    **Linda G. Shapiro**
University of Washington
{wisdomik,msaygin,fghezloo,dgeva,pka4,ranjay,shapiro}@cs.washington.edu
fatwir@uw.edu

## Supplementary material

We present the following items in the supplementary material section:

1. Data curation models, algorithms and parsing pipelines (Section A)
2. Exploratory analysis of the collected data (Section B)
3. Pretraining and downstream evaluation details (Section C)
4. Exploration of trained model representations (Section D)
5. A Datasheet [13] for our QUILT dataset (Section E)

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

System Prompt:
```
You are a histopathology text classifier
```

User Prompt:
```
Imagine you are a text classifier. Classify the given text into one of the following
surgical pathology types namely: Bone, Cardiac, Cytopathology, Dermatopathology,
Endocrine, Gastrointestinal, Genitourinary, Gynecologic, Head and Neck,
Hematopathology, Neuropathology, Ophthalmic, Pediatric, Pulmonary, Renal, Soft
tissue, Breast pathology. Output only the top 3 pathology types in an ordered python
list
```

Few-shot examples:
```
"Radicular cyst arises within the periodontal ligament space,
particularly the periapex from the epithelial cell of malassez. These
radicular cysts are caused by inflammation following the death of the
pulp extending into the periapical radix. Radicular cysts caused by
inflammation are always associated with a non vital tooth."

"['Soft tissue', 'Dermatopathology', 'Hematopathology']"
```

```
INPUT:
"There is a lesion with slight thickening of the muscularis mucosa and
submucosa. There is a subtle change in the lamina propria that doesn't
look quite like normal stromal cells. Description of slight thickening of
the muscularis mucosa and submucosa with subtle changes in the lamina
propria. Highlighted field shows the changes more dramatically. Abnormal
cells in the lamina propria that appear pink and spindly."

OUTPUT: "['Gastrointestinal', 'Soft tissue', 'Hematopathology']"
```

Figure 7: Prompting LLM with few-shot examples to extract the top three sub-pathology classification of a given text.

**Pre-processing multi-plane figures.** Many figures in academic papers are multi-plane, which means a number of sub-figures (Charts, graphs, histopathology and non-histopathology sub-figures) are placed next to each other to make a larger figure. We extracted individual images from multi-plane

figures to create multiple instance bags. To locate boundaries and white gaps between sub-figures, we utilized Sobel filters. Binary thresholding was then used to find the contours surrounding the sub-figures. We employ image size and image ratio thresholds to remove undesirable sub-figures and our histopathology classifier to maintain just histopathology sub-figures. We supply the histological sub-figures individually for this type of figure by appending "_[0-9]+" to the end of the multi-plane figure id. If a figure is divided into more than 5 sub-figures, we preserve the original image to ensure that the resolution of these sub-figures remains reasonable. Figure 8 shows an overview of this pre-processing step in different scenarios of successful and unsuccessful crops.

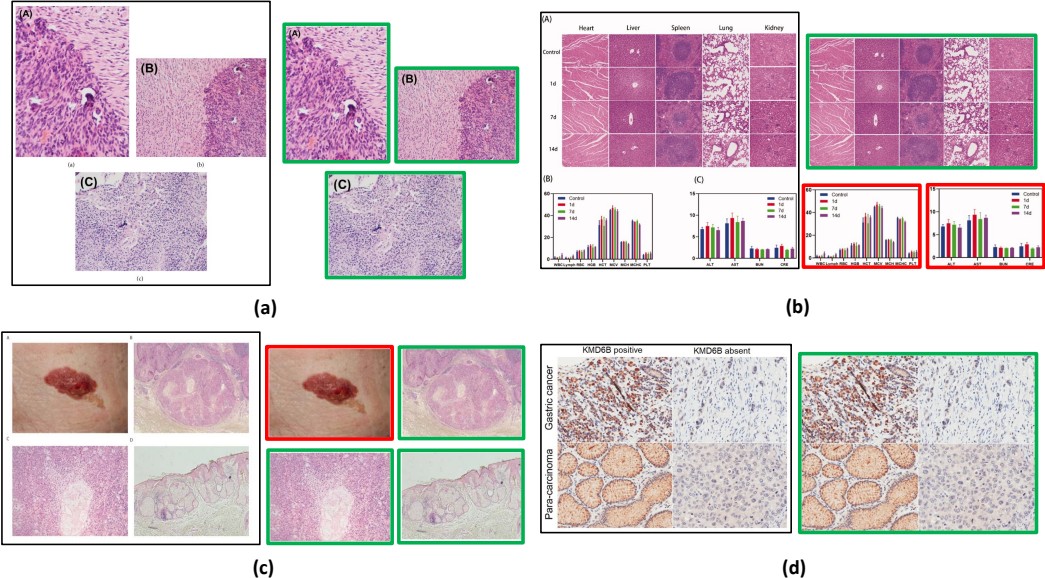

Figure 8: (a), (b), and (c) successfully cropped sub-figures where histopathology images (green box) are kept and non-histopathology (red box) images are removed. (b) histopathology crops are kept as not separated because the individual crops don't meet the size threshold so the original figure is kept. (d) Unsuccessful crop due to minimal gap between sub-figures. Original image is stored.

## A.5    Privacy preserving steps

In order to ensure privacy while handling the dataset, several steps were taken to protect sensitive information. These steps include:

- Reduction of Signal to Noise using a LLM: To protect the privacy of the dataset, a LLM was utilized to reduce the signal-to-noise ratio. By applying the LLM, irrelevant or sensitive information was masked or removed.

- Exclusion of Videos Not Fully in Narrative Style: Videos that did not adhere to a fully narrative style were intentionally left out of the dataset. This step was taken to minimize the risk of including any potentially private or sensitive content that could compromise individuals' privacy.

- Release of Video IDs and Reconstruction Code: Instead of directly releasing the complete dataset, only video IDs from YouTube were made public. Additionally, the code is provided to enable researchers to recreate the dataset.

- Collection from Diverse Channels: Data collection was performed from a wide range of sources, including both large and small channels. This approach aimed to decrease the risk of overfitting to specific channel types, thereby mitigating privacy concerns associated with potential biases linked to particular channels.

# B Exploratory analysis of the collected data

In this section, we provide the statistics of the QUILT dataset. Figure 4 illustrates the distribution of data across 18 sub-pathology types, offering a comprehensive analysis of the dataset's text distribution. Moreover, for additional statistical details regarding QUILT, please refer to Table 4, which presents supplementary information on various aspects of the dataset.

Table 4: Additional QUILT statistics

| Property | Average value |
| --- | --- |
| Medical text per image | 1.74 |
| ROI text per chunk | 2.30 |
| Medical text per chunk | 1.93 |
| Words per medical text | 22.92 |
| Words per ROI text | 8.75 |
| Images per chunk | 2.49 |
| Image-text pair per chunk | 2.36 |
| UMLS entity per medical text | 4.36 |
| UMLS entity per ROI text | 1.61 |

# C Pretraining and downstream evaluation details

## C.1 External Evaluation Datasets

**PatchCamelyon** Veeling et al. [37] contains 327,680 color images (96×96px) from histophathology scans of lymph node sections. The images are assigned a binary label indicating whether they contain metastatic tissue or not. **NCT-CRC-HE-100K** Kather et al. [20] consists of 100,000 non-overlapping image patches from hematoxylin and eosin (H&E) stained histological images (224x224px) of human colorectal cancer and is categorized into cancer and normal tissue. **SICAPv2** Silva-Rodríguez et al. [35] contains 182 prostate histology WSIs with 10,340 patches (512 x 512px) and both annotations of global Gleason scores and patch-level Gleason grades. Images are labeled as Non cancerous, Grade 3, Grade 4, and Grade 5. **Databiox** [6] consists of 922 Invasive Ductal Carcinoma cases of breast cancer. This data set has been collected from pathological biopsy samples of 150 patients which are labeled as Grade I, II and III. Each pathological sample in has four levels of magnification: 4x, 10x, 20x and 40x. **BACH** [4] consists of 400 WSIs of breast tissue which are labeled as normal, benign, in-situ and invasive carcinoma. **Osteo** [5] is a set of 1,144 patches (1024 x 1024px) taken from 40 WSIs representing the heterogeneity of osteosarcoma. Images are labeled as Viable tumor (VT), Non-tumor (NT) and Necrotic tumor (NEC). **RenalCell** [8] contains 27,849 images of clear-cell renal cell carcinoma H&E-stained (300 x 300px) annotated into five tissue texture types. **SkinCancer** [22] consists of 36,890 patches (395 x 395px) from WSIs skin biopsies from patients with Basal cell carcinoma (BCC), squamous cell carcinoma (SqCC), naevi and melanoma. Images were annotated for 16 categories: chondral tissue, dermis, elastosis, epidermis, hair follicle, skeletal muscle, necrosis, nerves, sebaceous glands, subcutis, eccrine glands, vessels, BCC, SqCC, naevi and melanoma. **MHIST** [38] contains 3,152 patches (224 x 224px) from 328 Formalin Fixed Paraffin-Embedded WSIs of colorectal polyps. These images are labeled as hyperplastic polyps (HPs) or sessile serrated adenomas (SSAs). **LC25000** [7] which we divide into **LC25000 (Lung)** with 15,000 and **LC25000 (Colon)** with 10,000 color images (768×768px). The lung subset is labeled as lung adenocarcinomas, lung squamous cell carcinomas, and benign lung tissues and the colon sebset is labeled as colon adenocarcinomas and benign colonic tissues. Table 5 summerizes these datasets.

Table 5: Downstream tasks and datasets. Note that SkinTumor dataset is a subset of SkinCancer. [μm per pixel - MPP]

| | Task | Sub-Pathology | Dataset | Classes | Magnification | Size (Train/-Val/Test) | Image-size |
|---|---|---|---|---|---|---|---|
| Classification | Lymph-node metastasis detection | Breast | PatchCamelyon [37] | 2 | 1 MPP | (0.75/0.125/0.125) * 327,680 | 96 x 96 |
| | Tissue Phenotyping | Colorectal | NCT-CRC-HE-100K [20] | 8 | 0.5 MPP | 89,434/ - /6333 | 224 x 224 |
| | Gleason scoring | Prostate | SICAPv2 [35] | 4 | 1 MPP | - / - /10,340 | 512 x 512 |
| | Bloom Richardson grading | Breast | Databiox [6] | 3 | [2,1,0.5,0.25] MPP | - / - /922 | (2100 × 1574), (1276 × 956) |
| | Tissue classification (normal, benign, in-situ and invasive carcinoma) | Breast | BACH [4] | 4 | 0.5 MPP | - / - / 400 | 2048 x 1536 |
| | Osteosarcoma classification (non-tumor, necrotic tumor, and viable tumor) | Bone | Osteo [5] | 3 | 1 MPP | - / - / 1,144 | 1024 x 1024 |
| | clear-cell renal cell carcinoma tissue phenotyping (renal cancer, normal, stromal, other textures) | Renal | RenalCell [8] | 5 | [0.5, 0.25] MPP | - / -/ 27,849 | 300 x 300 |
| | Classification of skin neoplasms and various anatomical compartments | Skin | SkinCancer [22] | 16 | .5 MPP | 28039/-/8851 [imb] | 395 x 395 |
| | Colorectal Polyp Classification | Colorectal | MHIST [38] | 2 | 1 MPP | - / -/ 3,152 | 224 x 224 |
| | Lung adenocarcinoma classification (normal, adenocarcinoma and SCC) | Lung | LC25000 (LUNG) [7] | 3 | - MPP | - / - / 15,000 | 768 x 768 |
| | Colon adenocarcinoma classification (normal and colon adenocarcinoma) | Colon | LC25000 (Colon) [7] | 2 | - MPP | - / - / 10,000 | 768 x 768 |
| Retrieval | histopathology image-text retrieval | - | Quilt-1M | 1.02M | - | 13,559 | - |
| | histopathology image-text retrieval | - | ARCH [12] | - | - | 7500 | - |

## C.2 QUILTNET Implementation

All model implementations in this study are built upon the open source repository OpenCLIP [16], which enables large-scale training with contrastive image-text supervision.The experiments were conducted using PyTorch and utilized up to 4 NVIDIA A40 GPUs. The hyperparameters for finetuning and training from scratch are provided in Table 6. During the training process, gradient checkpointing and automatic mixed precision (AMP) techniques were employed, with a datatype of bfloat16.

All models were trained with image size of 224, except for the finetuned ViT-B-32 models, where the images were first resized to 512 before randomly cropping them to the desired size of 224. In the case of ViT-B-32 finetuning, the data was kept stretched, meaning it maintained a one-to-one mapping between the image and text. However, for all other models, the data was unstretched. This means that for those models, sampling from medical texts occurred with a probability of $p = sample\ prob$, or sampling from ROI texts. Within the medical or ROI texts, sampling was done uniformly. For all finetuned GPT/77 models we use the OpenAI CLIP [30] pretrained network as initialization and for ViT-32 maintain the use of QuickGeLU[8]. We perform hyperparameter tuning for all linear

---

[8]https://github.com/openai/CLIP/blob/main/clip/model.py

probing results, exploring different values for learning rate, epochs, and weight decay. This process helped optimize the performance of the models during the linear probing stage.

Table 6: Training hyperparameters for QUILTNET

| Hyperparameter | Finetuning | Training |
|---|---|---|
| batch size (per gpu) | 256/1024 | 1024 |
| peak learning rate | 1e-5 | 5.0e-4 |
| learning rate schedule | constant | cosine decay |
| epochs | 15 | 40 |
| warmup (in steps) | 200 | 2000 |
| random seed | 0 | 0 |
| image mean | (0.48145466, 0.4578275, 0.40821073) | same |
| image std | (0.26862954, 0.26130258, 0.27577711) | same |
| augmentation | Resize; RandomCrop (0.8, 1.0) | RandomResizedCrop (0.8, 1.0) |
| optimizer momentum | $\beta_1, \beta_2 = 0.9, 0.98$ | same |
| weight decay | 0.1 | 0.2 |
| eps | 1.0e-6 | same |
| optimizer | AdamW [27] | same |
| $sample\ prob$ | 0.85 | same |

Table 7: Zero-shot image classification. accuracy (%). * denotes models trained from scratch. SkinTumor is the Neoplastic Subset of SkinCancer. Also note that PMB refers to PubmedBert, a BERT model of 256 context length pre-trained on PMC-15M. We swapped our model's text encoder from GPT2 to PubmedBert to assess performance differences

| | ViT-B/32 | | | | ViT-B/16 | | | |
|---|---|---|---|---|---|---|---|---|
| | CLIP | PLIP | QUILTNET | | CLIP | BiomedCLIP | QUILTNET | |
| Dataset | GPT/77 | GPT/77 | GPT/77 | (GPT/77)* | GPT/77 | PMB/256 | GPT/77 | PMB/256 |
| SkinCancer | 5.40 | 36.65 | **45.38** | 8.93 | 5.40 | 24.75 | 23.41 | **28.93** |
| SkinTumor | 10.35 | 56.36 | **58.29** | 36.26 | 13.85 | 37.0 | **51.47** | 51.20 |
| NCT-CRC | 26.4 | 54.02 | **59.56** | 17.35 | 20.09 | 51.71 | 28.68 | **59.20** |
| PatchCamelyon | 61.88 | 58.61 | **64.6** | 49.92 | 50.45 | 53.25 | **67.91** | 53.52 |
| MHIST | 52.92 | 57.52 | **62.54** | 44.52 | 52.3 | 40.23 | 44.32 | **52.71** |
| LC25000(LUNG) | 61.36 | 78.77 | **80.16** | 67.71 | 50.29 | 72.44 | 50.71 | **81.87** |
| LC25000(COLON) | 62.5 | 77.79 | **93.28** | 72.08 | 78.56 | **90.57** | 62.26 | 87.1 |
| SICAPv2 | 39.40 | **44.53** | 39.49 | 25.07 | 27.38 | **45.81** | 25.54 | 45.1 |
| BACH | 26.0 | **43.0** | 41.25 | 33.75 | 27.25 | 54.75 | 40.75 | **62.0** |
| Databiox | 37.53 | 39.48 | **42.52** | 32.32 | **33.51** | 31.24 | 33.19 | 29.93 |
| Osteo | 19.49 | 54.02 | **64.16** | 27.88 | 16.08 | 50.79 | 38.37 | **59.79** |
| RenalCell | 20.3 | 50.7 | **52.57** | 16.35 | 28.80 | 47.08 | 28.32 | **50.72** |

Table 8: Classes for each dataset on zero-shot image classification. Note that we used the same prompt templates for each dataset. The templates used are: ["a histopathology slide showing {c}", "histopathology image of {c}", "pathology tissue showing {c}", "presence of {c} tissue on image"]

| Dataset | Classes |
|---|---|
| SkinCancer | 'Necrosis', 'Skeletal muscle', 'Eccrine sweat glands', 'Vessels', 'Elastosis', 'Chondral tissue', 'Hair follicle', 'Epidermis', 'Nerves', 'Subcutis', 'Dermis', 'Sebaceous glands', 'Squamous-cell carcinoma', 'Melanoma in-situ', 'Basal-cell carcinoma', 'Naevus' |
| PatchCamelyon | 'Lymph node', 'Lymph node containing metastatic tumor tissue' |
| NCK-CRC | 'Adipose', 'Debris', 'Lymphocytes', 'Mucus', 'Smooth muscle', 'Normal colon mucosa', 'Cancer-associated stroma', 'Colorectal adenocarcinoma epithelium' |
| MHIST | 'Hyperplastic polyp', 'Sessile serrated adenoma' |
| LC25000Lung | 'Lung adenocarcinoma', 'Benign lung', 'Lung squamous cell carcinoma' |
| LC25000Colon | 'Colon adenocarcinoma', 'Benign colonic tissue' |
| BACH | 'Breast non-malignant benign tissue', 'Breast malignant in-situ carcinoma', 'Breast malignant invasive carcinoma', 'Breast normal breast tissue' |
| SICAPv2 | 'Benign glands', 'Atrophic dense glands', 'Cribriform ill-formed fused papillary patterns', 'Isolated nest cells without lumen rosetting patterns' |
| Databiox | 'Well differentiated bloom richardson grade one', 'Moderately differentiated bloom richardson grade two', 'Poorly differentiated grade three' |
| RenalCell | 'Red blood cells', 'Renal cancer', 'Normal tissue', 'Torn adipose necrotic tissue', 'Muscle fibrous stroma blood vessels' |
| Osteo | 'Normal non-tumor', 'Necrotic', 'Tumor' |
| SkinTumor | 'Squamous-cell carcinoma', 'Melanoma in-situ', 'Basal-cell carcinoma', 'Naevus' |

# D Exploration of trained model representations

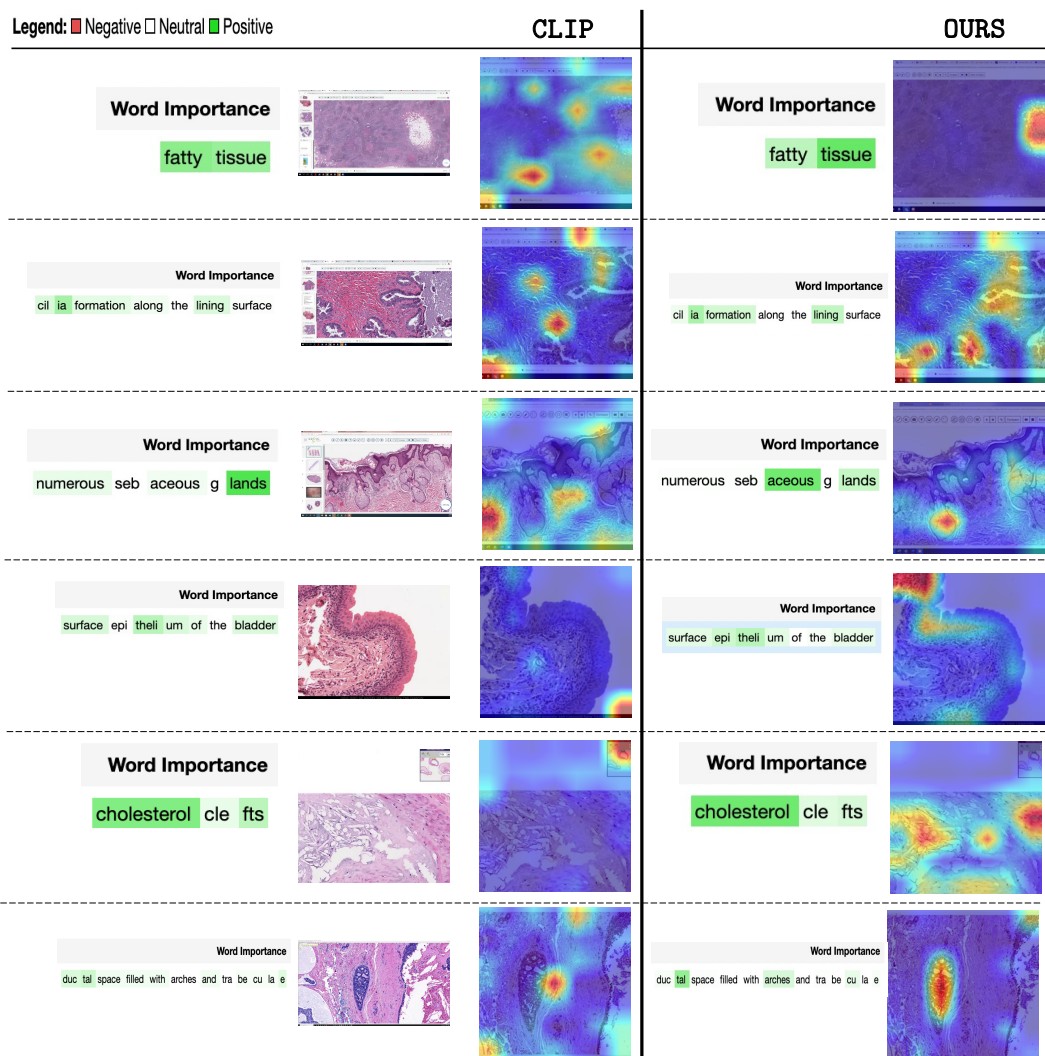

Figure 9: Comparison of the attention maps generated by QUILTNET and CLIP. The corresponding words are highlighted based on their importance. Attention masks were generated using GradCAM [34].

Table 9: UMAP visualization of image embeddings generated by QUILTNET from the different datasets listed in Table 5.

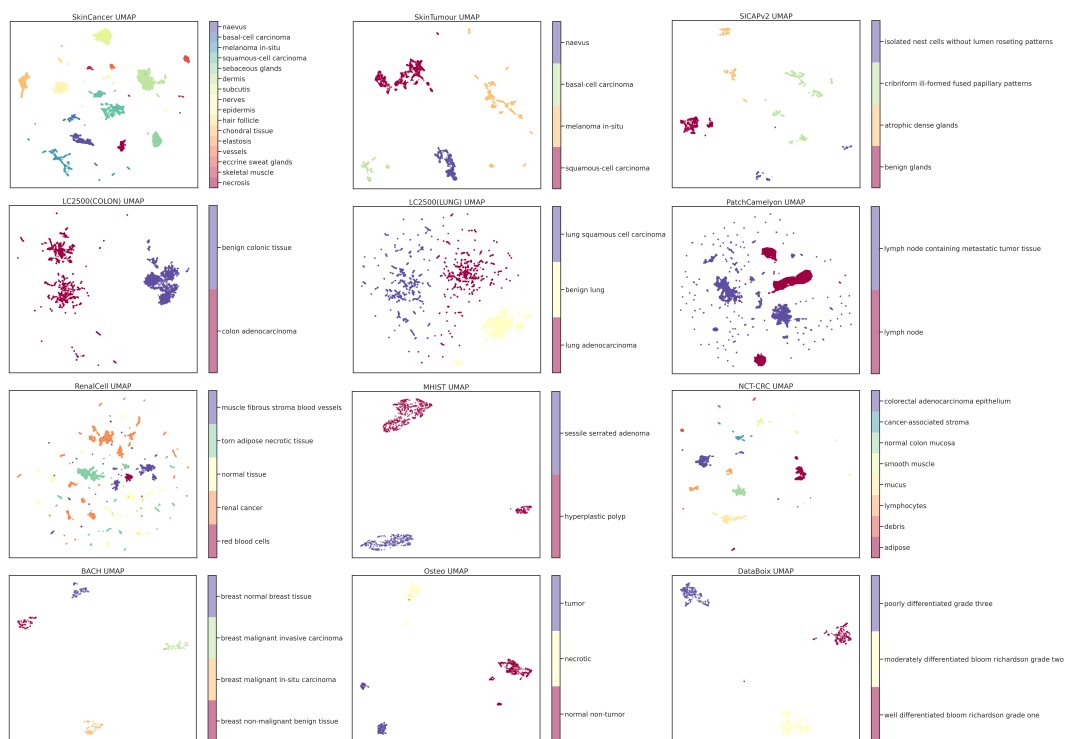

# E Datasheet for QUILT

In this section, we present a DataSheet [13] for QUILT, synthesizing many of the other analyses we performed in this paper.

1. Motivation For Datasheet Creation
   - **Why was the dataset created?** To train histopathology multi-modal models, on in-domain data, useful for diagnostically relevant downstream tasks.
   - **Has the dataset been used already?** Yes.
   - **What (other) tasks could the dataset be used for?** Could be used as training data for representation learning, and also for supervised learning on metadata
   - **Who funded dataset creation?** This work was funded by the Office of the Assistant Secretary of Defense328 for Health Affairs through the Melanoma Research Program under Awards No. W81XWH-20-1-0797329 and W81XWH-20-1-0798.

2. Data composition
   - **What are the instances?** The instances that we consider in this work are histopathology images derived from educational videos, paired with aligned text, derived from ASR and denoise using an LLM.
   - **How many instances are there?** We include greater than 1 million image-text pairs, from videos and additionally from less noisy sources like PubMed articles.
   - **What data does each instance consist of?** Each instance consists of an image, a descriptive text for the image as a whole and for its regions of interest, an estimated microscope magnification of the image, medical UMLS entities in the text, and the subpathology type. Each instance is representative of a video chunk based on where histopathology content is stable.
   - **Is there a label or target associated with each instance?** We use the raw ASR and LLM denoised captions as labels in this work as well as auxiliary information which includes magnification, UMLS entities and pathology type.

- **Is any information missing from individual instances?** Yes, for instances in the dataset that are not from QUILT (i.e videos), e.g. from PubMed Article datapoints, the additional auxiliary information is not included.

- **Are relationships between individual instances made explicit?** Not applicable – we do not study relationships between disparate videos (even from the same narrator) nor the relationship between chunks in the same video.

- **Does the dataset contain all possible instances or is it a sample?** Contains all instances our curation pipeline collected, as the list of videos is not exhaustive of what is available online, there is a high probability more instances can be collected in the future.

- **Are there recommended data splits (e.g., training, development/validation, testing)?** There are no recommended data splits, as this data was curated mainly for pretraining rather than evaluation.

- **Are there any errors, sources of noise, or redundancies in the dataset? If so, please provide a description.** Yes. Despite our numerous attempts to reduce noise using various models, algorithms and human knowledge databases, ASR is often noisy, and there are many erros that we cannot fix.

- **Is the dataset self-contained, or does it link to or otherwise rely on external resources (e.g., websites, tweets, other datasets)?** The dataset is self-contained. However, we plan to only release the video URLs and some paired non-pixel data points, rather than the videos themselves, so as to protect user privacy (allowing users to delete videos).

3. Collection Process

- **What mechanisms or procedures were used to collect the data?** We leveraged the YouTube API and the `youtube-dl` library.

- **How was the data associated with each instance acquired? Was the data directly observable (e.g., raw text, movie ratings), reported by subjects (e.g., survey responses), or indirectly inferred/derived from other data?** The data was directly observable (public) (from YouTube).

- **If the dataset is a sample from a larger set, what was the sampling strategy (e.g., deterministic, probabilistic with specific sampling probabilities)?** We used a probabilistic strategy with many algorithms and heuristics, more details are in Appendix A.1.

- **Who was involved in the data collection process (e.g., students, crowdworkers, contractors) and how were they compensated (e.g., how much were crowdworkers paid)?** Data collection was primarily done by the first authors of this paper.

- **Over what timeframe was the data collected? Does this timeframe match the creation timeframe of the data associated with the instances (e.g., recent crawl of old news articles)? If not, please describe the timeframe in which the data associated with the instances was created.** The data was collected from January 2023 to May 2023, even though the YouTube videos are often much older.

4. Data Preprocessing

- **Was any preprocessing/cleaning/labeling of the data done (e.g., discretization or bucketing, tokenization, part-of-speech tagging, SIFT feature extraction, removal of instances, processing of missing values)?** Yes, we discuss this in Section **??** and in Appendix A.1: of note, we use a large language model, UMLS database and a set of algorithms to 'denoise' ASR transcripts, an ensemble of histopathology classifiers to inform relevant segments of the video, and extract the representative image(s) for each video segment.

- **Was the "raw" data saved in addition to the preprocessed/cleaned/labeled data (e.g., to support unanticipated future uses)? If so, please provide a link or other access point to the 'raw' data.** The raw data was saved, but at this time we do not plan to release it directly due to copyright and privacy concerns.

- **Is the software used to preprocess/clean/label the instances available? If so, please provide a link or other access point.** Yes, software for downloading and processing the data is available on GitHub through our website.

- **Does this dataset collection/processing procedure achieve the motivation for creating the dataset stated in the first section of this datasheet? If not, what are the limitations?**
  Yes, the dataset does allow for the study of our goal, as it covers various histopathology sub-domains and provides crucial data points and metadata for pretraining. Some of its limitations we are aware of involve various biases on YouTube, as well as various inaccuracies of the models (e.g ASR model) within the curation pipeline, which we discuss in Appendix A.1 and A.3.

5. Dataset Distribution

- **How will the dataset be distributed?** We distribute all the derived data (denoised frames, captions, magnifications etc), as well as links to the YouTube videos that we used under the MIT license and strictly for research purposes.
- **When will the dataset be released/first distributed? What license (if any) is it distributed under?** The data has been released, under the permissible MIT license for research-based use only.
- **Are there any copyrights on the data?** We believe our use is 'fair use,' however, due to an abundance of caution, we will not be releasing any of the videos themselves.
- **Are there any fees or access restrictions?** No.

6. Dataset Maintenance

- **Who is supporting/hosting/maintaining the dataset?** The first authors of this paper.
- **Will the dataset be updated? If so, how often and by whom?** We do not plan to update it at this time.
- **Is there a repository to link to any/all papers/systems that use this dataset?** Not right now, but we encourage anyone who uses the dataset to cite our paper so it can be easily found.
- **If others want to extend/augment/build on this dataset, is there a mechanism for them to do so?** Not at this time.

7. Legal and Ethical Considerations

- **Were any ethical review processes conducted (e.g., by an institutional review board)?** No official processes were done, as our research is not on human subjects, however, because the dataset is in the medical domain we had significant internal discussions and deliberations when choosing the scraping strategy.
- **Does the dataset contain data that might be considered confidential?** No, we only use public videos.
- **Does the dataset contain data that, if viewed directly, might be offensive, insulting, threatening, or might otherwise cause anxiety? If so, please describe why** No – because many of these videos are medical and educational in nature, we have not seen any instance of offensive or abusive content.
- **Does the dataset relate to people?** Yes, it relates sometimes to deidentified patients, typically studied by pathologists.
- **Does the dataset identify any subpopulations (e.g., by age, gender)?** Not explicitly (e.g. through labels)
- **Is it possible to identify individuals (i.e., one or more natural persons), either directly or indirectly (i.e., in combination with other data) from the dataset?** Yes, some of our data includes content from known pathologists, albeit niche, they sometimes include their faces in the corner of the video. All of the videos that we use are of publicly available data, following the Terms of Service that users agreed to when uploading to YouTube.