# OpenReview forum: "Quilt-1M: One Million Image-Text Pairs for Histopathology"
_NeurIPS.cc/2023/Track/Datasets_and_Benchmarks — NeurIPS 2023 Datasets and Benchmarks Oral_

### Official Review · Reviewer_jVw5 · 2023-07-19
**Open-source multimodal dataset for Histopathology**

**Rating:** 8
**Confidence:** 4
**Correctness:** I don't have concerns about the corre…
**Clarity:** Yes

**Strengths:**

- The authors employed a novel approach to collect this dataset, integrating various state-of-the-art techniques. These include text extraction using ASR, text correction utilizing the LLM, and employing image classifiers to identify videos containing histopathology images.

- The distribution of all the codes used in constructing this dataset represents a substantial engineering endeavor.

- The release of this artifact holds significant importance for the research community. Recent scientific progress has underscored the value of comparable datasets containing numerous image and text pairs. However, existing sources of such datasets are often limited in size or not openly accessible (e.g., PMC-15M).



**Additional Feedback:**

Typo error on page 9, Discussion section: see A.6 in the Appendix (perhaps A.5). I didn’t find A.6 in the Appendix.

Typo error on page 2: 13 external histopathology datasets (I assume it’s 12 as you mentioned in section 4)

A question about dataset reproducibility:
- Can I expect a 100 % reproducible dataset? I’m asking this because videos can be deleted by users in the future for any reason


**Documentation:**

Yes

**Ethics:**

The authors have discussed this topic in the supplementary material (A.5).


**Limitations:**

- It would be interesting to see to what extent including different image-text sources such as LAION, Twitter, and PubMed can improve the accuracy of the proposed model. Could the authors conduct an ablation study to assess the impact of incorporating LAION, Twitter, and PubMed datasets on the QUILT dataset? This study would determine whether these additional sources are beneficial in enhancing the model's performance or if they potentially lead to a decrease in performance.

**Opportunities For Improvement:**

- In several parts of the papers, the authors state that QUILT and QUILT-1M do not overlap with existing data sources. My question is how did you check whether they overlap?
- Can authors provide reasons why training from scratch underperformed a fine-tuned CLIP model? Can it relate to some noisy transcriptions of images?



**Relation To Prior Work:**

The authors have well discussed the relationship of their work with the previous contributions.

**Summary And Contributions:**

The main contribution of this work is a multimodal image-text dataset comprising one million histopathology image-text pairs. The dataset encompasses various sources, including the newly created QUILT dataset, image-text pairs from PubMed open access, histopathology images from the LAION dataset, and Twitter data from OpenPath.

The authors thoroughly describe the collection process and provide insights into the properties of the resulting dataset. Furthermore, they introduce QUILTNET, a modified version of the CLIP model, and fine-tune it on their dataset. Subsequently, they assess its performance on 12 downstream histopathology datasets. Lastly, the paper extensively discusses technical limitations, safety considerations, ethical aspects, and potential biases associated with the research.

---

> ### Author Response · Authors · 2023-08-17
>
> We would like to appreciate reviewer jVw5 for their review comments on our study.
>
> **Reviewer jVw5 asked how the authors determined the data sources are disparate.**
> To clarify, we claim that QUILT does not overlap with existing data sources like LAION, OpenPath and PubMed, thus allowing us to merge it with those data sources to form QUILT-1M. QUILT does not overlap with existing histopathology image-text pair data (ARCH, OpenPath) because there is no overlap between the actual WSIs described between the two datasets.
>
> **Reviewer jVw5 asked why the finetuned CLIP model outperforms the CLIP model trained from random initialization (scratch).**
> We believe the underperformance of training from scratch is tied to empirical observations that models with CLIP's contrastive objective tend to excel with larger datasets when trained from scratch, as compared to other vision-language objectives, such as ITM in VILT or BEiT. This is especially evident when the text encoder is being trained from scratch.
>
> **Reviewer jVw5 suggested ablation experiments to determine the downstream impact of other sources of data on the QUILT data.**
> While we acknowledge that further ablation studies could provide deeper insights into the quality of each sub-dataset and how much they provide on top of QUILT, our academic computational constraints restricted the extent of our evaluations. Only large industry labs would be able to complete such experiments within a reasonable period of time. Thus we've designated this comprehensive analysis for future work.
>
> **Reviewer jVw5 identified typos.**
> Thanks for pointing out these typos! We have fixed the typo on page 9 in the discussion section, identifying the correct subsection in the Appendix as A5. We have corrected the number of external histopathology datasets to 13 (from 12) on page 2.
>
> **Reviewer jVw5 asked about the reproducibility of QUILT.**
> We take data reconstruction concerns seriously. While our dataset is largely reproducible, some slight differences may arise due to non-deterministic elements within our pipeline (e.g. median image wrt pixels), or as rightly pointed out by the reviewer, creators may take down videos. Hence, we have chosen to release the full dataset (QUILT-1M) restrictively to those who comply with user data privacy policies via Zenodo (https://zenodo.org/record/8239942) and GoogleDrive (https://forms.gle/TKohQ7zLwYfFn8qRA) after researchers agree to certain terms, protecting against further distribution of the dataset and committing to its specified research use.

---

> > ### Comment · Reviewer_jVw5 · 2023-08-29
> > **Thank you for the feedback**
> >
> > Thank you for your responses and also for making the dataset available.  I am happy with the feedback from the authors

---

### Official Review · Reviewer_rAj5 · 2023-07-20
**Review of QUILT-1M, an image-text histopathology dataset curated from YouTube and other sources**

**Rating:** 9
**Confidence:** 4
**Clarity:** The paper is well written.

**Strengths:**

Please find a list of strengths below:

1. The authors were clever to mine YouTube for this dataset. This is indeed a large resource, and I myself have watched several histopathology videos on YouTube.
2. The size of the QUILT dataset is a strength. The authors write that QUILT includes 419,780 images and 768,826 matched text pairs.
3. QUILT includes multiple sentences per image, which allows multiple descriptions of images.
4. The authors provide a detailed description of the method they used to create QUILT.
5. The experimental results (Section 4) are comprehensive.
6. Limitations and biases are discussed.
7. The code to create the dataset and models are publicly available. Thank you for hosting your models on HuggingFace. I am not affiliated with HuggingFace but I appreciate that your model is available for reuse. Please see point 11 in the “opportunities for improvement” for a related suggestion about the dataset. A critical weakness is that the final dataset is not publicly available.
8. The authors have created an effective website to describe the dataset.

**Additional Feedback:**

Great work overall. I have a few minor comments:

1. In line 3, the word “halted” is not precise. Research in this field has not been halted, though I would argue that research has not been at the same pace as in other fields. Please use a more precise phrase here.
2. In line 5, I would argue YouTube is not an “untapped” resource, though I am not familiar with any research in histopathology that uses data from YouTube. Please use a more precise phrase here.
3. In line 55-56, the authors write “... provide more expressive”. My question is, more expressive than what?
4. In line 94, the word “untapped” seems inappropriate, because in the previous sentences, the authors write that several studies have used YouTube videos.
5. In line 126, what is the resolution of the downloaded videos? The authors write “low-resolution”. Is a consistent resolution used? I wonder because the resolution of the downloaded videos presumably affects the resolution of the resulting images for QUILT.
6. Were any of the final image-text pairs checked for quality by a human? Section 3.3 (Quality) does not contain an answer to this.

**Correctness:**

The vast majority of the claims made are correct. I have noted claims that I do not agree with (please see 'Opportunities for improvement').

**Documentation:**

The authors provide sufficient detail on data collection and organization. However, the availability and maintenance are unclear to me. When I first read the manuscript, I was under the impression that the full (final) dataset was available. When I downloaded the CSV file however, it appeared that the download was intended to be used to re-create the dataset. It also seems that the CSV can only be used to recreate QUILT but not QUILT-1M. It is not clear whether QUILT-1M is shared publicly. If the authors are not able to share the final dataset, they should make this clear in the manuscript.

Regarding maintenance, I suggest that the authors host their dataset on a platform that assigns a DOI, like Zenodo or HuggingFace. This will alleviate my concerns of persistence and findability.

In addition, the manuscript does not assign a license to the dataset. Can the authors please include this information in the manuscript?

**Ethics:**

I do not suspect any ethical concerns.

**Limitations:**

The authors adequately address limitations and social biases.

**Opportunities For Improvement:**

Please find a list of opportunities for improvement below:

1. It appears to me that the QUILT and QUILT-1M datasets are not publicly available in their final form. What is the reason for this? This is a critical weakness in my opinion and is one of the main contributors to my score. (Please also see item 11 for suggestions of hosting the dataset.) If the dataset cannot be shared publicly, the authors should indicate this clearly in the manuscript and include steps that users must take to regenerate the dataset. This is particularly a concern for the subset of QUILT-1M curated from Twitter, because Twitter is no longer publicly viewable without an account and its API has been rate limited, making the re-creation of QUILT-1M much more difficult.
2. In the manuscript, the authors do not provide the license under which they distribute QUILT and QUILT-1M. Please specify the applicable license.
3. How are the authors certain that the narrators of the YouTube videos are “experts? The quality of the textual descriptions relies on the expertise of the narrator.
4. I disagree with the authors’ statement on lines 62 and 63. Patch-based classification, region segmentation, nuclear segmentation, and other tasks have all been commonly done with machine learning for many years now. Weakly-supervised learning is just one of many methods in computational pathology.
5. In lines 64 and 65, the authors claim that weakly-supervised models are “sub-optimal” and cite two papers. Neither of these references provides evidence for this claim. In fact, I would argue that the model proposed in reference 11 (the MCAT model) is far from “sub-optimal”.
6. Please describe the scale of the histopathology images earlier on in the manuscript. This is described in lines 250-251 but I would argue that it is important to mention early on (perhaps somewhere in the introduction).
7. Please identify which LLMs are used.
8. Please specify the resolution of the resulting image dataset. What is the average height and width of the images?
9. In line 261, please specify which pre-trained model was used. What was it pretrained on?
10. Near line 226 (“Twitter Data from OpenPath”), please describe in more detail how your approach differs from that of OpenPath. The difference is not clear to me. A short description of the OpenPath dataset would likely address this concern.
11. I urge the authors to host the dataset on another platform other than Google Drive, because programmatic download (e.g., from the command line or scripts) can be difficult from Google Drive. In fact, I have never succeeded in downloading a file from Google Drive programmatically, and for this reason, I suggest that the authors host QUILT and QUILT-1M on another platform. I highly recommend hosting the dataset on the HuggingFace Hub or on Zenodo. In both cases, a DOI can be created for the dataset which is useful in identifying the source of the data. This improves findability and reusability.
12. The authors provide a CSV of the video IDs and text. This CSV contains 802,186 rows (not including the header). The authors write that QUILT contains 768,826 image-text pairs, and QUILT-1M contains around one million images. Where does the number 802,186 come from? Related to this, is this CSV enough to reconstruct QUILT-1M? Or is this only to construct QUILT? This is unclear to me.
13. Related to 12 above, is the GitHub repository sufficient to recreate QUILT-1M? After reviewing the code, I believe it generates QUILT but not QUILT-1M. Is this correct? If so, how does one access QUILT-1M?

**Relation To Prior Work:**

The authors discuss how their work differs from previous work.

**Summary And Contributions:**

The authors present QUILT-1M, a dataset of image-text pairs of histopathology images. Many of these images and textual descriptions are curated from YouTube as well as Twitter and other sources. This is a clever method to mine histopathology images with textual descriptions and follows a few recent papers [1, 2]. This dataset is rich, and the authors demonstrate this with assessments of zero-shot, full-shot, and linear probing. A major contribution of this dataset is its size -- the authors claim it is the largest dataset of its kind to date. To accomplish this scale, the authors used automated tools, like large language models, to aid in dataset creation.

My main concerns are of dataset findability and accessibility. The QUILT dataset (the subset of QUILT-1M sourced from YouTube) seems to not be available in its entirety. The authors have provided a CSV with URLs of YouTube videos used in QUILT and code to recreate the dataset (and in fact another user has experienced problems recreating this dataset: https://github.com/wisdomikezogwo/quilt1m/issues/7). In addition, I am not able to locate the full QUILT-1M dataset. The GitHub repository https://github.com/wisdomikezogwo/quilt1m appears to generate QUILT but not QUILT-1M. My concerns of dataset findability and accessibility are the main contributors to my score of 5. I am glad to improve this score if (1) the authors make the QUILT and QUILT-1M datasets fully available or (2) the authors provide a rationale for not releasing the final QUILT and QUILT-1M datasets. If the authors are not able to make the datasets available, I strongly suggest that they revise the manuscript to reflect the fact that users will have to re-generate the dataset in order to use it.

References
1. Huang, Z., Bianchi, F., Yuksekgonul, M., Montine, T., & Zou, J. (2023). Leveraging medical Twitter to build a visual–language foundation model for pathology AI. bioRxiv, 2023-03.
2. Lu, M. Y., Chen, B., Zhang, A., Williamson, D. F., Chen, R. J., Ding, T., ... & Mahmood, F. (2023). Visual Language Pretrained Multiple Instance Zero-Shot Transfer for Histopathology Images. In Proceedings of the IEEE/CVF Conference on Computer Vision and Pattern Recognition (pp. 19764-19775).

---

> ### Author Response · Authors · 2023-08-17
>
> We highly appreciate reviewer rAj5 for their comments on our work.
>
> **Reviewer rAj5 asked if the dataset can be made available directly due to difficulty in reconstruction.**
> We have chosen to release the full dataset (QUILT-1M) restrictively to those who comply with user data privacy policies via Zenodo (https://zenodo.org/record/8239942) and GoogleDrive (https://forms.gle/TKohQ7zLwYfFn8qRA) after researchers agree to certain terms, protecting against further distribution of the dataset and committing to its specified research use. Users can refer to the links on the website and the GitHub repo to request access to QUILT-1M.
>
> **Reviewer rAj5 asked for the manuscript to specify the applicable license.**
> We have added this to the manuscript under the Datasheet for Dataset section. The data is shared under the MIT license (as described in our Repository and on openreview.)
>
> **Reviewer rAj5 asked how the identified narrators were considered/vetted to be experts.**
> Histopathology, being a specialized field, ensures that the majority of those discussing it possess notable expertise. Given the limited number of channels from which we curate content (only 278 in total), we undertook a manual assessment to verify the credibility of our narrators. Our findings revealed that most of these channels belong to hospitals or specialized doctors. We identified channels that had an educational format, and/or aimed their content towards an audience of other experts, implying a certain level of proficiency. Thus, even without direct assessment of each narrator’s capabilities, we theorize that our narrators are not only well-informed but are professionals with the confidence to disseminate educational content to their peers on public platforms.
>
> **Reviewer rAj5 suggested changes to certain claims made in the manuscript regarding previous methods in computational pathology.**
> We acknowledge the reviewer's point. As outlined in lines 62-64 of the paper, we highlight the prevailing trend in computational pathology to operate at the WSI level. This is primarily driven by the costs associated with data collection at the patch level. It's crucial to clarify, however, that our assertion doesn't suggest the absolute nonexistence of any other CPATH task beyond weak-supervision on the slide level. Our use of "suboptimal" is to emphasize that models trained at the WSI level tend to underperform on patch-level tasks, particularly because they often rely on embeddings from out-of-distribution feature extractors, such as unfinetuned ResNets. We will make these statements clearer.
>
> **Reviewer rAj5 identified omissions of contextual information on models used and some dataset statistics in the manuscript.**
> We have made the needed changes; reporting the LLM version used in the main paper (line 41) as GPT 3.5 and in the main paper as well as the appendix, the average resolution (height and width) of images, which is 882 x 1468 pixels (line 253), the scale of images (line 37), and the pre-trained model leveraged as OpenAI pre-trained CLIP model (line 264).
>
> **Reviewer rAj5 asked for a detailed data acquisition description of how our approach differs from OpenPath.**
> With OpenPath releasing only the tweet IDs and associated text, leaving out images and the post-publication changes to Twitter's API, which became more restrictive and costly, our approach involved manual data curation complemented by our custom Twitter crawlers. Any discrepancies between our version of OpenPath and the original can be attributed to manual curation challenges and potential tweet deletions over time. It's essential to clarify that we solely utilized the tweet IDs shared by OpenPath without any additional inclusions.
>
> **Reviewer rAj5 suggested the use of dataset-hosting platforms with easy programmatic  access for hosting QUILT-1M.**
> To aid findability, reproducibility, and most especially to provide the data to researchers without the CPU compute hours to run the reconstruction code, we offer access to the resized images version of QUILT via Zenodo (https://zenodo.org/record/8239942) and the full-sized images via GoogleDrive (https://forms.gle/TKohQ7zLwYfFn8qRA) due to restrictions in upload-size on other platforms.

---

> ### Author Response · Authors · 2023-08-17
>
>
> **Reviewer rAj5 asked about the discrepancy in the manuscript-reported size of QUILT vs the observed size in the released CSV.**
> We apologize for the oversight. The correct size of QUILT (which is the largest part of QUILT-1M) is actually 802,148 image-text pairs from the possible 802,186 rows in the released CSV, 38 of which had no captions. The figure of 768,826 mentioned in the paper is incorrect. Additionally, the actual number of images is 437,878, not the 419,780 stated in the paper. This discrepancy occurred because we accidentally shared only the training set while excluding the holdout set in our initial report. Also, the official number of the others are incorrect as they were numbers before quality checks, specifically, our PubMed subset has 59,371 pairs down from 62,458 pairs, our Laion subset has 22,682 pairs down from 23,240 pairs, and our OpenPath subset has 133,511 down from 133,526 reported in the manuscript, hence totaling up to 1,017,712 samples. The manuscript has been corrected to reflect this.
>
> **Reviewer rAj5 asked about access to the complete QUILT-1M, as the reconstruction code released is specific to QUILT and not any other data source that makes up QUILT-1M.**
> Indeed, our code/pipeline is specifically designed to reproduce only QUILT, not QUILT1M. To obtain QUILT1M, one would either
> need to extract histology images from sources like Pubmed, Laion, and Openpath individually or
> directly request access to the dataset through Zenodo (https://zenodo.org/record/8239942) and GoogleDrive (https://forms.gle/TKohQ7zLwYfFn8qRA). These links are also available in the GitHub repo and website of the project.
>
> **Reviewer rAj5 suggested changes to certain claims and choice of words in the manuscript.**
> We agree with the reviewer and have changed the word from "halted" to “slowed” in the abstract (line 3).
> Prior to our work Youtube or any online video platform has not been leveraged as a source of histopathology image and text data for training ML models, hence our use of the word untapped in line 5.
> Finally, the reviewer is correct in noticing the word “untapped” is unfit for line 94. We have omitted the word ‘untapped’ from the manuscript at that line.
>
> **Reviewer rAj5 asked for the reasoning behind the use of the word “expressive” on lines 55-56.**
> “Expressive” here is used to mean textual descriptions that clearly describe the image features and the underlying histologic concepts e.g describing the shape of a cell as well as the underlying consequences of that cell, as compared to text from say articles' figure captions that aren’t as descriptive due to either many factors e.g page limits, descriptions unaligned with the figure.
>
> **Reviewer rAj5 asked about the resolution of downloaded videos that make up QUILT.**
> Low resolution simply means we download the lowest possible resolution a Youtube video has to provide, this is variable but typically is around 320p, so we set 320p as a minimum resolution used for search ONLY newly referenced on line 126. For the image extraction that forms the dataset we use the highest possible resolution, typically 1080p, and have referenced this on line 254. We’ve added the average height and width of the images on line 253 and this provides a sense of the average resolutions of videos dowxnloaded.
>
> **Reviewer rAj5 asked if any image-text pairs were checked for quality by humans.**
> Due to the extensive size of the data, our quality control measures were confined to eliminating non-histopathology images (considering our histopathology classifier has an approximate 5% false positive rate) and removing histopathology images accompanied by non-descriptive texts from all datasets (including PubMed, QUILT, Laion, and OpenPath). Nevertheless, the superior downstream performance of our model indirectly attests to the accurate alignment and quality of the data.

---

> > ### Comment · Reviewer_rAj5 · 2023-08-21
> > **Thank you**
> >
> > Thank you for your detailed responses and also for making the dataset available. I have raised my score to 9.

---

### Official Review · Reviewer_7iJR · 2023-07-21
**Quilt-1M: One Million Image-Text Pairs for Histopathology**

**Rating:** 8
**Confidence:** 5
**Clarity:** [Yes.]

**Strengths:**

- The paper is well written and easy to follow and motivates the need for such a dataset in coherent manner.

- The authors target histopathology images, this is very relevant and timely. As opposed to many existing vision-language models in medical literature that focused on different modalities, the curation of this dataset tailored to pathology is truly a commendable effort.

- The use of Youtube videos is novel and interesting.  The internet is considerably an unlimited source of information, and the designed curation procedure considers several aspects from filtering to modalities alignment after text-extraction from frames.

- The authors extensively evaluated the curated dataset via downstream tasks after pre-training, achieving state of the art performance on several tasks i.e., zero/few-shot and cross modal retrieval.

**Additional Feedback:**

[]

**Correctness:**

The dataset is constructed in a structured and careful manner. The authors detailed each curation strategy.

**Documentation:**

[Yes]. URLs and documentation are included for better understanding.

**Ethics:**

Public sources were employed for curation, and the research was not on human subjects.

**Limitations:**

- It is unclear to what extent human experts verified the curated text-image pairs and the level of involvement i.e., how many expert pathologists confirmed the false positive rates.

- A CLIP model was pre-trained on the 1M dataset, did the authors evaluate the potential of the fine-tuned model on downstream tasks when trained with less data (<< 1M). This may be useful to assess the utility of this dataset in low-resource environments (hardware etc.).

**Opportunities For Improvement:**

[See Limitations]

**Relation To Prior Work:**

The discussed works are sufficient.

**Summary And Contributions:**

This work introduces a large-scale vision language dataset for histopathology consisting of 1 million image-text pairs curated from public sources including YouTube videos, Twitter, and research papers. To curate such a dataset, several automated processes are employed i.e., mixture of models, language models, handcrafted algorithms, and automatic speech recognition in order to collect more representative videos, filter narratives, and extract text efficiently. As opposed to existing vision language datasets for pathology, this work is more comprehensive in terms of collections and identifies potential biases herein, proving to be a truly large-scale dataset that can enable further research in multi-modal representation learning for histopathology. To validate the utility of the curated dataset, several text-based metrics are employed, along with a pre-trained CLIP model for downstream tasks i.e., zero-shot and linear probing across several datasets with diverse sub-pathologies.

---

> ### Author Response · Authors · 2023-08-17
>
> We thank reviewer 7iJR and appreciate the time and effort invested in reviewing our work.
>
> **Reviewer 7iJR asked if any experts verified the data pairs and their level of involvement.**
> No extra expert-pathologist’s hours were spent verifying the data. It's important to underscore that our data already originates from medical experts, implying an inherent level of accuracy and credibility. While we recognize concerns about the post-processing phase and potential inaccuracies therein, we are cognizant of this. We have verified a small subset for inaccuracies manually and are actively investigating potential inaccuracies as we build upon this contribution in our next project.
>
> **Reviewer 7iJR asked if ablations with varying dataset sizes were carried out to further extrapolate the dataset's utility in low-resource environments.**
> Firstly, it's crucial to emphasize that we released multiple QuiltNet models on Huggingface of varying model capacities (three models to be exact, ViT-B-32|GPT77, ViT-B-16|GPT77, and ViT-B-16|PMB-256). This ensures that researchers with limited resources can access and benefit from our models without having to undertake extensive training themselves.
>
> Second, training a single model demanded significant computational power and time, using up 4-6 A100 GPUs for 12 hours for training. The resource intensity of the process substantially limited our ability to conduct additional ablations. Furthermore, in resource-scarce scenarios, considerations frequently pivot towards model size and FLOPs, more so given the extensive size of our dataset. This influenced our decision to train VIT models using varied patch sizes and text tower configurations, ensuring consistent dataset use across tests.
>
> Finally, we concede that a more granular analysis examining the correlation between dataset size and training performance would be beneficial. However, owing to the aforementioned computational limitations, we refrained from such an analysis. That said, both Figure 3 and Table 1 offer indirect evidence of the advantages conferred by a more substantial training dataset. Our model's performance, when juxtaposed with PLIP [1] (trained on a smaller dataset), is distinctly superior.
>
> In light of these points, we commit to executing the suggested ablations and updating the manuscript in subsequent iterations.
>
> - [1] Huang, Zhi, et al. "Leveraging medical Twitter to build a visual–language foundation model for pathology AI." bioRxiv (2023): 2023-03.

---

> > ### Comment · Reviewer_7iJR · 2023-08-25
> > **Post rebuttal**
> >
> > I would like thank the authors for the responses and subsequent updates to the manuscript. Having read other reviews, I am convinced this work would be valuable to community and inspire more extensions for foundation models specific to histopathology. Overall, I am happy to raise my initial score.

---

### Official Review · Reviewer_8biP · 2023-07-21
**Review comments**

**Rating:** 10
**Confidence:** 5
**Correctness:** Yes.
**Clarity:** A very well written paper.

**Strengths:**

The paper is well written. The dataset generation process is well described, which ensures the quality of the dataset. Compared to prior histopathology text-image datasets, QUILT-1M contains more sample pairs. The limitation and potential social ethic issues are also discussed. I have no doubt that the QUILT-1M dataset and the multimodal model QUILTNET have significant contribution to digital histopathology.

**Additional Feedback:**

I was a bit surprised to find the underperformance of QUILTNET on BACH and SICAPV2. Particularly, BACH is a widely used histopathology image classification dataset. But zero-shot using QUILTNET is 14% worse than BiomedClip. Is there any comments about this?

**Documentation:**

Yes, very detailed information on data collection.

**Ethics:**

I think there might be a minor ethics concern. But since the dataset is collected from Youtube and public domain, the tissue images may contain subject privacy information. So I flag this paper and leave it to ACs for a final decision.

**Limitations:**

Yes, the authors discussed the limitations of the dataset due to the involvement of automation tools for data collection. In addition, the study is also aware of the social bias in data collection process.

**Opportunities For Improvement:**

1. It would be greatly appreciated if the authors could make the QUILT or QUILT-1M dataset accessible via Google Drive or other cloud platforms. Although the toolkits to regenerate the dataset are available, there might be numerous challenges to address during the reconstruction process, as outlined in the paper. These challenges could potentially result in a lower-quality dataset if regenerated.

2. As the authors have pointed out, the QUILT-1M dataset exclusively comprises image-text pairs with English text, potentially introducing social biases during model training. I know it is difficult and challenging, but I do wish the dataset would expand and include more diverse data.

**Relation To Prior Work:**

Yes. All of the quality and sample amount of QUILT-1M and the resulting QUILTNET model differentiate this study from previous works.

**Summary And Contributions:**

Extensive multimodal models have demonstrated considerable promise for various applications. Nonetheless, the absence of a high-quality image-text histopathology dataset has impeded the use of such large multimodal models in digital histopathology. To address this issue, this paper bridges the gap by introducing the QUILT-1M text-image dataset. Furthermore, the authors fine-tune the CLIP model using the QUILT-1M dataset, leading to histopathology-specific CLIP-Similar model called QUILTNET. The availability of both the dataset and the multi-modal model is expected to significantly enhance research in digital histopathology.

---

> ### Author Response · Authors · 2023-08-17
>
> We would like to thank this reviewer 8biP for their comments to improve our submission.
>
> **Reviewer 8biP suggested that QUILT(-1M) be made public due to the reconstruction challenges.**
> While our dataset is largely reproducible, some slight differences may arise due to non-deterministic elements within our pipeline e.g. (median image w.r.t pixels). Hence, we have chosen to release the full dataset (QUILT-1M) restrictively via Zenodo (https://zenodo.org/record/8239942) and GoogleDrive (https://forms.gle/TKohQ7zLwYfFn8qRA) after researchers agree to certain terms, protecting against further distribution of the dataset and committing to its specified research use.
>
> **Reviewer 8biP suggested an expansion of QUILT(-1M), including data modalities and multi-lingual captions.**
> We're grateful to the reviewer for highlighting the importance of dataset diversity, a concern we wholeheartedly share and prioritize. To streamline our initial efforts, we focused exclusively on the English language for two primary reasons: 1) It facilitates a straightforward evaluation of LLM outputs, and 2) It maintains consistency with prior work [1], which solely comprises English-written tweets. However, as part of our ongoing endeavors, we are in the process of curating an expanded medical dataset from YouTube. This new collection will encompass various medical domains beyond histopathology and will include content in multiple languages.
>
> **Reviewer 8biP had some ethical concerns with the source of the images narrated by the experts which make up QUILT(-1M).**
> The dataset is sourced from videos presented by medical experts who are already well-acquainted with the imperatives of protecting Protected Health Information (PHI), in compliance with privacy rules and laws, with histology images in the public domain from platforms such as pathpresenter and histologyguide.com. We limit the curation of our dataset to only publicly available YouTube videos which have already scraped private patient information from the histopathology images being discussed.
>
> **Reviewer 8biP had asked about the underperformance of QuiltNet on BACH in comparison to BiomedClip.**
> We hypothesize that the dataset's limited size and our choice of text prompt might play a role in this.  Additionally, breast tissue is among the underrepresented sub-pathologies within QUILT. This might explain the underperformance on BACH, in which the distribution shift may be more pronounced than other breast-related datasets. Finally, BiomedClip [2] training benefits from more samples (possibly more medical breast-related samples).
>
> - [1] Huang, Zhi, et al. "Leveraging medical Twitter to build a visual–language foundation model for pathology AI." bioRxiv (2023): 2023-03.
> - [2] Zhang, Sheng, et al. "Large-scale domain-specific pretraining for biomedical vision-language processing." arXiv preprint arXiv:2303.00915 (2023).

---

> > ### Comment · Reviewer_8biP · 2023-08-20
> > **Thanks for your feedback to my comments.**
> >
> > I am happy with the feedbacks from the authors. Considering the substantial workload in this study, I really hope this work could be accepted.

---

### Official Review · Reviewer_MJNv · 2023-07-22
**Excellent contribution with a vast, information-rich histopathology dataset. However, concerns about copyright and data quality need addressing.**

**Rating:** 7
**Confidence:** 3
**Correctness:** No correctness issue.

**Strengths:**

1. The paper addresses the scarcity of analogous data in the medical field, specifically in histopathology, which has halted comparable progress. The introduction of QUILT-1M can help overcome this limitation and enable researchers to develop better models for histopathology. The dataset can have significant implications for the field of histopathology, which can ultimately benefit patients by enabling better diagnosis and treatment.

2. This paper provide a comprehensive pipeline about the data preprocessing and post-processing with the a mixture of models, including large language models, handcrafted algorithms, human knowledge databases, and automatic speech recognition, to curate the datase. This information is pretty helpful for the research community to create more open-sourced datasets in other domains.

**Additional Feedback:**

N/A

**Clarity:**

Overall, the paper is well-written. However, the authors should provide further clarification on a couple of points:

1. The relationship and differences between CLIP and QuiltNet need to be more explicitly defined. From the presented information, it appears that different encoders, such as VIT-B/32 and VIT-B/16 for image encoding and GPT-2 for text encoding, are used. If the architectures differ, it remains unclear how the authors were able to train QuiltNet by fine-tuning a pre-trained CLIP model.

2. The concept of 'few-shot performance with linear probing' requires additional explanation. Specifically, does 'shot' refer to the sample-level or the class-level?

3. In section 3.3, what is the definition of "conditioned" and "unconditioned"?

**Documentation:**

Authors provide a sufficient details on data collection and organization but do not mention the ethical and responsible use.

**Ethics:**

Considering the dataset is licensed by the MIT and it could be used by commercial medical institutions, the data quality remains questionable as the authors discussed in the section 3.3 and section 5.

Besides, the copyright issue should be clarified because the data is collected from the YouTube.

**Limitations:**

While the authors did not sufficiently address the limitations, it's somewhat understandable given the limitations of existing tools in comprehending and interpreting medical terminology within the histopathology domain. Further data refinement could potentially benefit from the inclusion of expert human input.

Additionally, based on existing literature, training CLIP tends to require much larger datasets compared to other pretraining architectures. I would recommend exploring other self-supervised algorithms such as ViLT and BEiT. Pretraining these models from scratch with the proposed dataset could potentially yield superior results compared to simply fine-tuning the pretrained CLIP model.

**Opportunities For Improvement:**

Further evaluation and comparison are necessary to determine the quality of the data in relation to other related datasets. Given the substantial scale of the data, examining each sample individually presents a challenge. However, an alternative could be to pre-train the same model on different datasets and evaluate their performance using identical testing data. In this paper, the authors exclusively demonstrate the results of models trained with their proposed Quilt dataset, without comparison to other datasets like ARCH and OpenPath.

**Relation To Prior Work:**

Yes. Briefly speaking, the authors introduce QUILT-1M, the largest vision-language histopathology dataset with 1M image-text samples, far surpassing previous datasets. They acknowledge the difficulty in discerning histopathology-relevant portions in broader datasets like PMC-15M. This paper enhances the field by tackling the limitations of existing medical datasets, particularly in histopathology.

**Summary And Contributions:**

The paper presents QUILT-1M, a large vision-language dataset for histopathology with a million image-text pairs. It's created using a blend of language models, algorithms, knowledge databases, and speech recognition. Using QUILT-1M, a fine-tuned CLIP model surpasses existing models in zero-shot, linear probing tasks, and cross-modal retrieval tasks for classifying various histopathology images.

---

> ### Author Response · Authors · 2023-08-17
>
> We thank this reviewer for their suggestions to improve the paper.
>
> **Reviewer MJNv asked if data quality can be evaluated by pretraining the same network on QUILT(-1M) vs other datasets and measuring performance on a fixed test set as a proxy to measure data quality.**
> While we acknowledge the need for an extensive evaluation to determine the quality of our dataset, our current assessments already illustrate the distinct advantage QUILT offers. This is particularly evident when contrasting the performance of our VIT-32-B model with the  PLIP VIT-32-B model [1] that was trained on OpenPath [1]. Given that the core architecture of both models remains unchanged, with the dataset being the sole differing factor, any performance disparities underscore the enhanced quality of our data relative to OpenPath. Additionally, the public version of ARCH [2], comprising 7.5k entries, is notably smaller than OpenPath, being lesser by over an order of magnitude. Hence, we opted to employ ARCH as an external test set, focusing on measuring our model's zero-shot performance on it.
>
> **Reviewer MJNv suggested that we acknowledge a few more limitations.**
> First, we will add a statement that further human refinement might improve the quality of the dataset. Second, we concur with the reviewer's perspective that alternative models, such as BEIT or VILT, might potentially deliver superior performance given the volume of data we possess. Nonetheless, to minimize our computational overhead rather than training with discrete data subsets like OpenPath, we maintained consistency by using an architecture aligned with PLIP [1], ensuring straightforward performance evaluations. We will incorporate this point into our limitations section, recognizing it as an avenue for future exploration. In fact, we are already in the process of leveraging this dataset with varied learning objectives.
>
> - [1] Huang, Zhi, et al. "Leveraging medical Twitter to build a visual–language foundation model for pathology AI." bioRxiv (2023): 2023-03.
> - [2] Gamper, Jevgenij, and Nasir Rajpoot. "Multiple instance captioning: Learning representations from histopathology textbooks and articles." Proceedings of the IEEE/CVF conference on computer vision and pattern recognition. 2021.

---

### Decision · Program_Chairs · 2023-09-22

**Decision:**

Accept (Oral)

**Comment:**

The paper presents a multimodal image-text dataset comprising one million histopathology image-text pairs, highlighting innovative data collection methods involving ASR and LLM for text extraction and correction. While concerns were raised regarding dataset accessibility, overlap with existing sources, and the performance gap between fine-tuning and training from scratch, the reviewers were highly enthusiastic about the merits of this work, including the comprehensive distribution of code, the dataset's significance for the research community due to its size, clear and well-written content, and ethical considerations. I believe this work will be of significant value to the research community.